# Restoration cannot be scaled up globally to save reefs from loss and degradation

**Clelia Mulà[1], Corey J. A. Bradshaw [2,3,4], Mar Cabeza [1], Federica Manca [1], Simone Montano[5,6] & Giovanni Strona [7]** ✉

Coral restoration is gaining popularity as part of a continuum of approaches addressing the widespread, recurring mass mortality events of corals that—together with elevated and chronic mortality, slower growth and recruitment failure—threaten the persistence of coral reefs worldwide. However, the monetary costs associated with broad-scale coral restoration are massive, making widespread implementation challenging, especially with the lack of coordinated and ecologically informed planning. By combining a comprehensive dataset documenting the success of coral restoration with current and forecasted environmental, ecological and climate data, we highlight how such a coordinated and ecologically informed approach is not forthcoming, despite the extent of previous and ongoing efforts. We show that: (1) restoration sites tend to be disproportionally close to human settlements and therefore more vulnerable to local anthropogenic impacts; (2) the immediate outcomes of restoration do not appear to be influenced by relevant ecological and environmental predictors such as cumulative impact; and (3) most restored localities have a high and severe bleaching risk by the middle of this century, with more than half of recently restored sites already affected. Our findings highlight the need for the coral reef community to reinforce joint development of restoration guidelines that go beyond local objectives, with attention to ocean warming trends and their long-term impacts on coral resilience and restoration success.

Coral reefs are among the most diverse ecosystems worldwide, yielding shelter and nourishment for >30% of named marine species[1] while simultaneously providing livelihoods, food security and protection from storms and coastal erosion to almost 1 billion people[2]. As coral reefs transform in the Anthropocene, their traditional roles and functions will also probably change. The benefits we receive from coral reefs will increasingly depend on the interactions between social and ecological systems and the emergence of novel service configurations

resulting from changing environmental and social dynamics[3]. However, coral reefs are increasingly threatened by climate change and local human stressors such as overfishing and pollution[4]. Reducing these local impacts can improve reef resilience and recovery following climate-induced bleaching events. For example, reefs with reduced land–sea impacts in Hawai'i demonstrated a greater probability of maintaining reef-builder cover (hard coral and crustose coralline algae) after marine heatwaves[5]. Models based on different scenarios of future

[1]Faculty of Biological and Environmental Sciences, Organismal and Evolutionary Biology Research Programme, University of Helsinki, Helsinki, Finland. [2]Australian Research Council Centre of Excellence for Australian Biodiversity and Heritage, Wollongong, New South Wales, Australia. [3]Australian Research Council Centre of Excellence for Indigenous and Environmental Histories and Futures, Cairns, Queensland, Australia. [4]Global Ecology, Partuyarta Ngadluku Wardli Kuu, College of Science and Engineering, Flinders University, Adelaide, South Australia, Australia. [5]Department of Earth and Environmental Sciences, University of Milano-Bicocca, Milan, Italy. [6]Marine Research and High Education Center, Magoodhoo Island, Republic of Maldives. [7]European Commission, Joint Research Centre, Ispra, Italy. ✉e-mail: giovanni.strona@ec.europa.eu

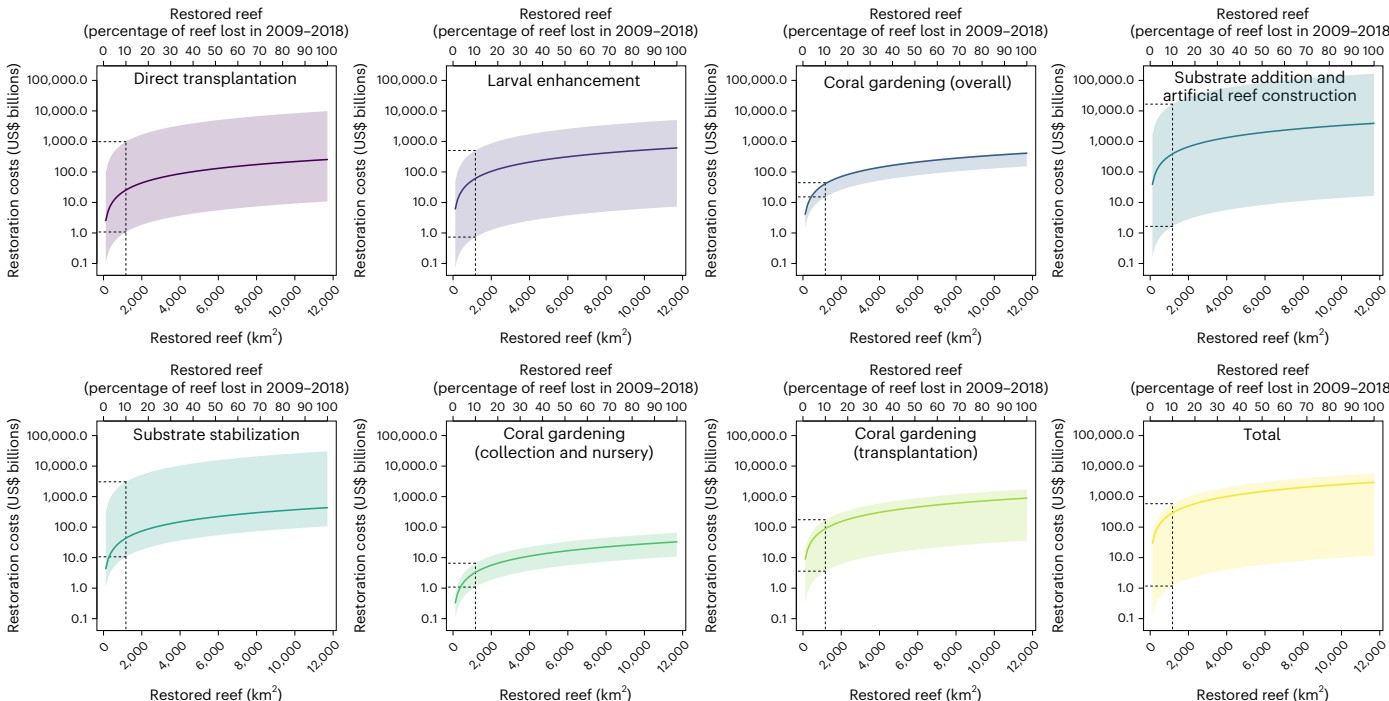

**Fig. 1 | Quantifying the hypothetical cost of broad-scale coral restoration.** The plots show the upper and lower bounds (shaded areas) and median values (solid lines) of estimated restoration costs of coral reefs based on published values as a function of the area of degraded reef that could be rehabilitated according to different techniques. The data used to estimate the upper, median and lower boundaries are from Table 1 of Bayraktarov et al.[32]. We report these data together with estimates of the costs to restore 10% of the 11,700 km² (that is, 1,170 km² = 117,000 ha) area of damaged reef (dashed lines in each plot).

carbon emissions point to major losses of coral cover globally by the middle of this century[6–8]. Predicting the responses of reefs to future disturbance is complicated by the variability of coral responses to thermal stress. While selective mortality of more vulnerable coral species can modify the resilience of coral communities under certain conditions[9], the general trend shows that repeated stresses might progressively erode the ecological resilience of remaining reefs[4,10].

Coral restoration is advocated as a promising tool to reduce coral loss and restore damaged reefs[11–16]. The US National Academies of Sciences, Engineering, and Medicine lists 23 intervention types to mitigate coral loss, including assisted gene flow, evolution and colonization, cryopreservation and microbiome manipulation[17]. The most commonly used restoration methods include outplanting coral colonies (with or without a nursery phase), corallivore removal, fish reintroductions to enhance ecosystem functioning, substratum enhancement and construction of artificial reefs[18–21]. Other techniques include releasing reared or harvested coral larvae[22,23], which promotes genetic diversity within coral populations[24] and potentially increases ecosystem resilience.

However, 30–40% of coral reef restoration projects fail due to poor planning, unrealistic objectives, inadequate regular maintenance and persistent anthropogenic pressures[21]. This high failure rate also includes trial-and-error projects designed to test and improve restoration practices[16]. These experimental projects advance restoration ecology by refining techniques, identifying challenges early and experimentally testing new methods. Although such trials might not yield immediate restoration success, they provide direction to improve future projects and contribute to understanding the risks and limitations of different approaches[16]. Early attempts suggest that restoration can succeed at the local scale (200 m² to 2 ha) within months to years[12,25–27]. There are emerging examples of successful restoration of reef function at broader scales (≤8 ha). For instance, coral restoration can drive rapid recovery of reef carbonate budgets following disturbances[28,29]. Although reef function can be restored at these scales,

the broader applicability and future scalability of these outcomes still need examination[30].

Even in ideal situations where restoration is deemed successful, only a small portion of the coral cover already lost can be realistically restored when comparing the costs with the pace of loss. Although restoration costs are readily available, quantifying any associated benefits is complex and conditional on long-term success. Coral restoration, even for only a few target species, is expensive[30–33], ranging from US$6,000 to US$143 million per hectare, depending on the environmental settings, techniques applied and local cost of labour[32]. This cost range includes coral gardening (coral fragments grown in nurseries transplanted back to the reef), direct transplantation of coral fragments, artificial reef structures, larval enhancement and low-tech methods such as transplanting coral fragments onto natural substrata in low-energy (currents or waves) environments[32] (Extended Data Table 1). However, these cost estimates do not include advanced techniques, such as assisted evolution or gene flow applicable to only a few species. Their costs and scalability remain highly uncertain because they have not yet been tested at scale[30]. Including these techniques would probably increase the total costs substantially. Some lower-cost examples do exist, such as tourism-led coral planting on the Great Barrier Reef, which leverages existing infrastructure to reduce costs to US$2.34 per coral per trip[5,34–36].

Despite these efforts, mass mortality events triggered by ocean warming and exacerbated by other local stressors can rapidly destroy huge areas of coral reef[37]. Quantifying the global loss of coral area is challenging. One study suggested that ~11,700 km² of coral reefs (14% of total coral coverage globally) was degraded from 2009–2018[38]. Even at the lower boundary of cost estimates and assuming all restoration actions are successful, rehabilitating just 10% of degraded coral reef areas would require >US$1 billion (Fig. 1 and Extended Data Table 1)—nearly four times the total investment in coral restoration over the past decade (US$258 million)[31]. At the upper boundary, costs could soar to US$16.7 trillion (Fig. 1 and Extended Data Table 1)—surpassing

the entire 2024 US research and development budget (US$210 billion)[39]. More economical options, such as coral gardening, are estimated to cost ~US$3.3 billion (Fig. 1 and Extended Data Table 1). More intensive approaches (for example, adding substratum for artificial reefs) could drive costs up appreciably. These figures highlight the vast monetary commitment required—ranging from US$6,000 to US$261 million per hectare—within a reasonable time frame.

In contrast to public perception[40–42] and scientific enthusiasm[40,43,44], these estimates show that restoration alone is not a practical or affordable solution to counteract the global decline of coral reefs. The challenge extends beyond monetary constraints; growing and transplanting billions of corals are formidable hurdles, as recognized previously[16,21]. This does not imply that restoration is ineffective at the local scale and under specific conditions, but it casts doubt on its scalability and role for coral conservation. The growing focus on restoration interventions also risks diverting attention from addressing the root causes of coral reef decline, such as reducing greenhouse gas emissions and mitigating local human land–sea impacts, and fostering optimism that might exceed the current evidence for effectiveness[45]. This calls for the identification of well-defined and realistic targets, such as the conservation of sites and/or species[46–48] selected according to criteria informed by ecological, economic and social considerations[49,50].

Such criteria should ideally maximize the success of restoration, but defining success is a challenge in itself[51,52] due to the lack of consensus on how to measure and quantify the economic and socio-cultural values of ecosystem outcomes[50,53–57]. Most studies[21] measure success in terms of survival and/or the growth of restored corals over a certain monitoring time[58]. This seemingly straightforward measure might be deceptive because of variable and often insufficiently long monitoring, and because the survival of the transplanted corals might not be representative of the overall condition of the reef, including habitat complexity, structural integrity and diversity[59]. For example, there is evidence that reefs with higher habitat complexity (for example, with abundant branching corals) offer more shelter and resources for fish and invertebrates, which in turn promote faster coral recovery following disturbances[59]. Another common metric is the percentage of live coral cover[60]. However, some studies do not incorporate a control and a reference baseline in restoration plans[61]. Without a comparison with nearby damaged yet unmanipulated reefs and establishing baselines, distinguishing natural recovery processes from restoration success is confounded.

In principle, these measures capture important and complementary facets of restoration success, which when combined offer standard metrics that are useful for assessing, selecting and improving restoration techniques[15,47,48,62–65]. However, despite the calls for standardized data collection protocols[51], this potential is hampered by the large variation in how monitoring data are collected, stored, processed and shared[21].

An additional complication in evaluating restoration success is that short-term, local ecological success (that is, growth and survival) might not necessarily increase the long-term persistence of restored reef ecosystems[48,63] and associated marine life[66,67]. The immediate results derived from coral interventions might only be temporary, especially in environmental settings facing persistent adverse conditions. For example, one study described the initial success of a coral restoration intervention in Indonesia, but then reported the death of almost all corals only 6 months after the first assessment[37]. Many restoration projects lack comprehensive monitoring frameworks to track long-term success. Even large, transnational corporations claiming leadership roles in ecosystem restoration often do not report the outcomes of their restorations[68]. This highlights the need for increased transparency, consistency and accountability in coral restoration to ensure that projects deliver meaningful and lasting ecological benefits.

We used data from 220 coral restoration projects globally (Extended Data Fig. 1) to answer the following questions: (1) what ecological and environmental factors are associated with the location of restoration projects? (2) are the same factors associated with restoration success or is success more associated with the features of restoration practice or local threats? and (3) will restored sites remain viable in the future? For the scope of this study and based on the information in available datasets, we define success as survival relative to monitoring time. We argue that the lack of consistent and clear targets and standardized, well-defined and ecologically meaningful metrics of success complicate planning and monitoring, thereby increasing the risk of wasting resources on projects that are unnecessary and/or doomed from the start. These weaknesses also slow progress by making it difficult to derive valuable information from past and ongoing projects to improve future endeavours. Focusing on coral gardening (with a nursery phase) and direct transplantation as the most common restoration techniques, we combined a large dataset of coral restoration actions with ecological, environmental and climate data to explore these questions quantitatively. We hypothesized that some of these variables (that is, remoteness from large human settlements, coral diversity, cumulative impact and bleaching events) are associated with both site selection and restoration success. For example, highly impacted sites might be more degraded, warranting restoration[69]. Additionally, sites more vulnerable to impacts face higher risks of conservation interventions being jeopardized by external stressors, influencing their restoration success or long-term viability.

## Results

### Criteria for restoration site selection

We explored whether coral sites targeted for restoration share common ecological and environmental attributes indicating how restoration sites are selected. We included variables associated with threats (climate change, pollution and exploitation) and practical implementation and biodiversity values (Extended Data Fig. 2). These included pre-restoration exposure to thermal anomalies, particularly the trend and mean values of bleaching alert levels from the National Oceanic and Atmospheric Administration (NOAA) Coral Reef Watch[70] in the five years preceding the target restoration (Methods) and the mean and trend of the cumulative impact on oceans of 14 stressors from human activities and climate change (that is, commercial demersal destructive fishing, commercial demersal non-destructive high- and low-bycatch fishing, pelagic high- and low-bycatch fishing, artisanal fishing, sea surface temperature increase, ocean acidification, sea level rise, shipping, nutrient pollution, chemical pollution, direct human damage and light)[71]. We also considered remoteness, measured as the travel distance from large human settlements[72], and gravity (combining remoteness with human population density)[73]. These two measures are related, but not redundant—remoteness provides a standardized measure quantifying the difficulty of accessing a reef (that is, a proxy for the logistical challenges) and gravity is a standardized measure of potential human–reef interactions[72]. We also took local coral diversity into account because it could potentially drive the choice of restoration sites by contributing to their perceived conservation value[74].

We built boosted regression tree models that predicted the choice of restoration sites with reasonable accuracy (mean false positive error ± s.d. = 0.25 ± 0.17; average false negative error = 0.27 ± 0.22; average true skill statistic = 0.48 ± 0.18; $n$ = 1,000 iterations; Methods). Gravity was the most influential variable, followed by mean cumulative impacts, coral diversity and remoteness (Fig. 2). We investigated the direction of the effects of the predictors using partial dependency plots. These revealed that the probability of a site being selected for restoration increased sharply with human accessibility to reefs (increasing gravity and decreasing remoteness), increased with impact and decreased with coral diversity (Fig. 2). In other words, restoration actions are more likely to occur in easily accessible localities subject to local human influences and possibly in a more degraded state (with lower coral diversity) than more remote, less accessible and less impacted localities[72].

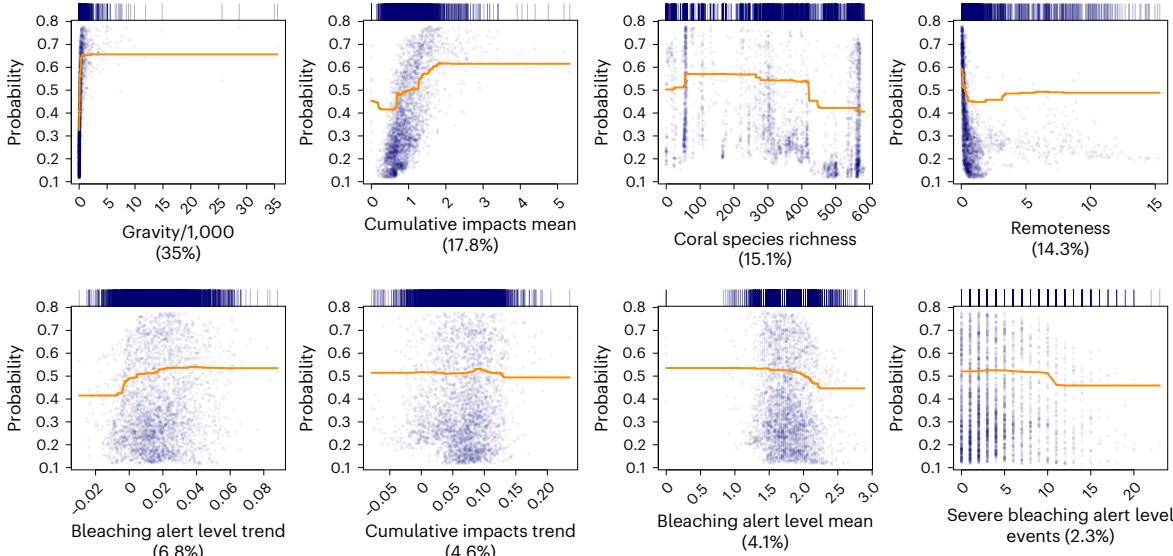

**Fig. 2 | Factors affecting the choice of restoration sites.** Partial dependency plots showing the marginal effects of the independent variables included in our model on the predicted probability of a site being the target of restoration. We report both the fitted functions (solid orange lines; $n = 10,000$) and fitted values (blue dots; $n = 3,324$). The rug plots on top of the panels show the density of observed values in the target independent variables. The percentage values in parentheses indicate the relative influence of each variable. Gravity was obtained from ref. 73 and represents the summed gravity of locations within a radius of 500 km from a target reef location, with individual gravity of each location within the radius computed as the ratio between the population of that location, and the squared distance (measured as travel time, in minutes) from the location and the target reef location. In the plot, we report the gravity values quoted by 1,000 to ease visualization of labels in the $x$ axis. For the quantification of cumulative impacts, refer to ref. 71. Remoteness is reported as log-transformed travel time from a target reef location to the closest large human settlement, as in ref. 72. Bleaching alert levels refer to NOAA Coral Reef Watch data[70] and we considered as 'severe bleaching alert' level events, those with alert levels I or II (Methods).

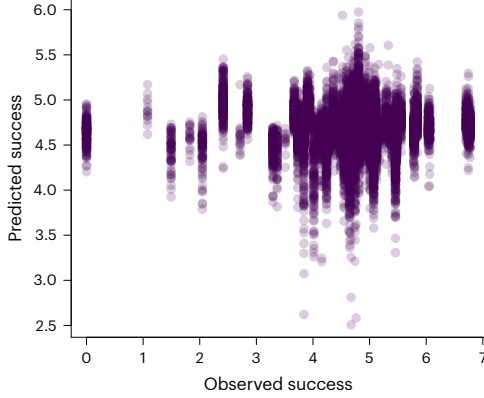

**Fig. 3 | Observed success of coral restoration projects compared with the success predicted by 1,000 boosted regression tree models based on a large set of independent variables identifying coral restoration techniques, ecological factors and disturbances.** The data are from a cross-validation exercise in which we trained and tested 1,000 models on spatially independent sets of observations (including 80% of data in the training sets and 20% in the testing sets). The mean $R^2$ of observed versus predicted restoration success in the 1,000 replicates was $0.05 \pm 0.06$ s.d. and the overall $R^2$ for all of the points in the plot was 0.00002. See Methods for details on how 'success' was quantified.

## Correlates of short-term coral restoration success

To assess drivers of restoration success, we generated boosted regression tree models including the restoration technique(s) (coral gardening, direct transplantation, artificial reefs, larval enhancements and number of coral genera used in restoration), remoteness, gravity, local coral diversity, mean and trend of cumulative impacts and pre- and post-exposure (±5 years) to thermal anomalies as independent variables (Extended Data Fig. 3).

Despite eight independent variables and parameter tuning, the models explained <5% of the variance in restoration success (assessed through cross-validation (Fig. 3); see also the Methods section 'Determinants of short-term restoration success' and Extended Data Fig. 4 for how we assessed success). This low performance could arise because of the few records in the dataset (134 after excluding the records for which we could not compute standardized survival; see Methods). However, it might also reflect broader issues, such as inconsistencies in monitoring and reporting across studies—an indicator of the lack of coordination in global restoration (see Discussion).

## Post-restoration fate of target sites

We explored how often the restored sites experienced severe bleaching following restoration. Sea surface temperature (bleaching alert levels from NOAA combining instantaneous heat stress and degree heating weeks (DHW); see Methods)[75] indicated that in 57% (170/299) of the localities we considered, coral communities were exposed at least once to a bleaching alert level I or worse (instantaneous heat stress > 1 °C and DHW = 4–8 °C-weeks) within five years following restoration (Fig. 4), with most localities experiencing multiple exposure events (mean ± s.d. = 1.54 ± 0.94). Fifteen percent of restored sites were exposed at least once to bleaching alert level II (instantaneous heat stress > 1 °C and DHW > 8 °C-weeks), with a mean of 1.3 ± 0.65 events per reef.

We then explored the potential fate of restored reefs by examining projected future heat stress. We referred to an intermediate greenhouse gas emissions scenario (Shared Socioeconomic Pathway 2-4.5 (SSP2-4.5))[76,77] under which the world does not shift substantially from historical social, economic and technological patterns—thus presenting moderate mitigation and adaptation and an associated radiative forcing of 4.5 W m⁻². We used data from ref. 78 to identify all of the years when the maximum projected DHW value was ≥20 °C-weeks for each locality (that is, near-complete mortality in >80% of corals[79]; Methods). Almost all (96.7%) reef localities are predicted to experience at least one year with a DHW value of ≥20 °C-weeks by 2100 (Figs. 5 and 6

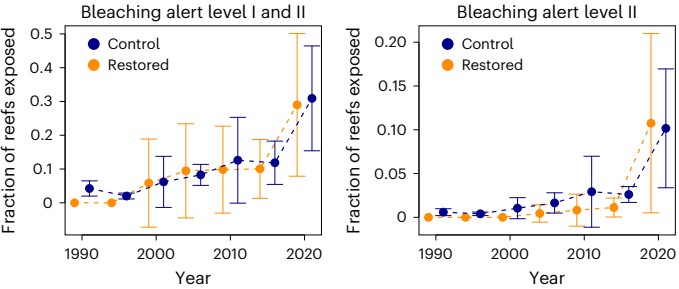

**Fig. 4 | Fraction of recently restored sites exposed to bleaching alert level I and/or II per year compared with control (non-restored) sites.** For each target year, we identified all of the localities restored in the preceding five years and identified as a control all of the remaining reef localities. We then computed the fraction of restored and control localities that were exposed to bleaching alert levels I and/or II. The data are aggregated in 5-year intervals, with values on the x axis indicating the upper boundary of the interval (that is, 1990 = 1986–1990). We shifted the time series for the restored and control data horizontally by ±1 unit to ease visualization. The data are presented as means ± s.d. ($n = 5$).

and Extended Data Fig. 5). Restored reefs are predicted to be the most vulnerable; we predict that 99.6% of them will experience at least one bleaching event—and more often many of them—by 2100 (24.8 ± 14.9 events), with the first anomaly recorded in 2063 on average (±16 years).

## Discussion

Our boosted regression tree models revealed clear predictors for the selection of restoration sites that were also correlated with success. Reef accessibility was a main determinant indicating how the local availability of infrastructure and facilities, targets of funding schemes and governmental or political actions often influence conservation planning more than broader ecological considerations[80–83]. For example, coral diversity had minimal influence in the choice of restoration sites, despite diversity being a good indicator of ecosystem resilience[84] and a metric that should be included when selecting restoration sites[15,48]. Many initiatives arise locally, driven by immediate needs and socio-cultural priorities, leading to patterns that might lack a clear ecological rationale. Proximity to human settlements, while offering socio-ecological benefits, is associated with greater degradation and human influence. This raises the question of whether it is better to focus restoration on a site where the need is moderate but the likelihood of success is high or a site with high restoration need but low chances of success[85]. While locally driven restoration can enhance socio-ecological resilience, contribute to livelihoods and protect coastlines, these efforts might be unsustainable and ineffective in the long term. Alternative interventions directly targeting these needs might be a safer and more effective strategy. Although social and cultural dimensions are beyond the scope of our study, we recognize their importance for restoration success and emphasize the need for ecologically informed guidelines that integrate local objectives.

In contrast, we could not identify clear, consistent factors correlated with restoration success. Under the hypothesis that major environmental and ecological conditions affect the success of restoration, the lack of consistency in the definition of success weakened the statistical power. For instance, the monitoring duration varied from 0.8–144 months, complicating comparisons of coral survival and therefore restoration success. However, we cannot exclude the notion that other unmeasured variables (for example, site-specific management practices, local disturbances and unforeseen ecological interactions) also determined restoration success. The varying spatial resolutions of our predictors also probably influenced the lack of relationships. For example, predictors such as coarse-scale sea surface temperature might not align well with the local conditions at restoration sites, in contrast with more localized measurements

such as the distance from population centres. The large variation in restoration techniques and environmental settings adds further complexity, making it difficult to generalize about the overarching drivers of success. Differences in the coral species targeted and the extent of local anthropogenic impacts also probably contributed to the poor performance of the model, complicating the identification of predictors of success.

The lack of common, standardized and widely accepted recommendations and protocols for data collection, processing and reporting[51] not only makes it challenging to link restoration success to specific restoration practices or environmental and ecological settings; they also have detrimental consequences for planning and management[30]. To document restoration actions in line with our findings, there are a few fundamental metrics emerging from our analyses. These include a detailed characterization of the restored area, ideally incorporating relevant environmental and ecological parameters to establish a baseline for evaluating restoration outcomes. For coral gardening initiatives, recording data such as the number of transplanted colonies per species, as well as their survival and growth at the target and adjacent control sites, would be valuable. Regular assessments over a standard period would enhance the reliability of these metrics. Quantifying human intervention through comparable measures such as person-hours or estimated costs for activities such as structure maintenance and algae removal would also aid in assessing resource requirements. We acknowledge that the extent of monitoring is contingent on the objectives of each restoration programme and local resource availability. We focus here on highlighting essential metrics that would improve the comparability and effectiveness of restoration at a broader scale. In line with recent guidance documents[86], our suggestions are not exhaustive, but they do emphasize the data needed to streamline monitoring and improve standardization.

The restoration process could benefit from initiatives that establish a single, centralized, open-access database hosting consistent data reported by scientists and practitioners following the establishment of common, globally relevant guidelines. This might be an extension of existing resources, such as the one we used here[21] (now hosted at www.icriforum.org/restoration/coral-restoration-database), or a new product, but it should provide a widely accepted template with which to plan data collection in future restoration actions. Although coral gardening is one such technique that could be included, restoration and rehabilitation practices rely on many techniques. The database should therefore accommodate data from diverse methodologies to provide a resource for all types of coral restoration. Successful

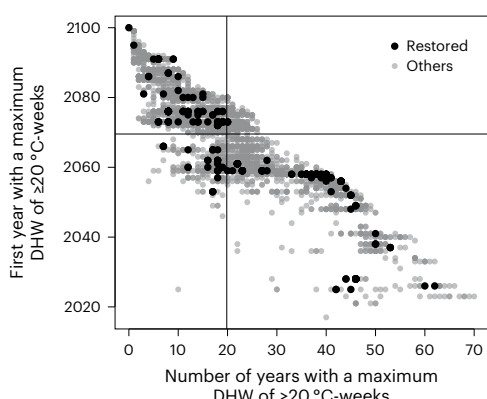

**Fig. 5 | Number of predicted end-of-century coral mass mortality years compared with the timing of the first mass mortality year under an intermediate emissions scenario (SSP2-4.5).** The black dots indicate restored localities and the grey dots represent the other reef localities that have not undergone restoration (0.5° × 0.5° grid cells). Mass mortality years for a given locality are identified as those when the maximum projected DHW value is ≥20 °C-weeks. This analysis was based on projected DHW data from ref. 78. The vertical and horizontal solid lines indicate the means of the x and y values.

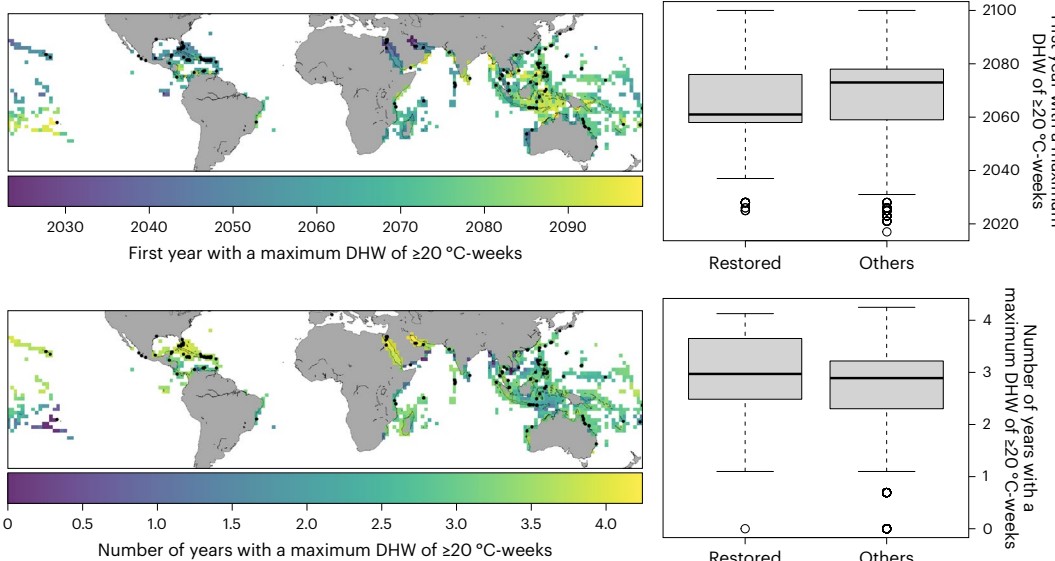

**Fig. 6 | First occurrence and frequency of years when reef localities (*n* = 3,929) experienced a maximum DHW value of ≥20 °C-weeks under an intermediate emissions scenario (SSP2-4.5).** Restored sites (*n* = 256) are identified in the maps by small black dots within the coloured pixels. The projected DHW data are from ref. 78. The boxplots summarize the values shown in the maps, with boxes indicating first and third quartiles, horizontal lines indicating median values, whiskers indicating the largest and lowest points inside the range defined by the first or third quartile +1.5 times the interquartile range and circles indicating outliers.

examples of similar databases include the MERMAID platform for reef monitoring (www.datamermaid.org) as an effective model for data standardization and sharing in marine science. Accessibility is essential for ensuring widespread use of and engagement with such a database. To maximize inclusivity, the resource should be openly accessible, user friendly and equipped with multilingual interfaces. By doing so, we can support diverse stakeholders involved in restoration planning and implementation, empowering them to contribute to and benefit from coral restoration around the globe.

Our analyses also show that restoration planning so far has underestimated the exposure of coral reefs to climate change[87], despite rising temperatures being the main driver of coral losses[8]. More than 30% of restored sites have already experienced high heat exposure shortly after restoration (Fig. 4) and all of them are projected to experience several more events of thermal stress in the coming decades (Figs. 5 and 6). Focusing on the short-term success of restoration could therefore be deceptive. Although not necessarily a revelation for some in the restoration community, it underscores the need for a more careful and strategic approach to restoration. Spending effort to restore an area with a high exposure to heatwaves could waste finite resources[88]. This risk might be reduced by considering future climate projections in restoration planning[89] to identify areas expected to face moderate stress, also potentially serving as recruitment hubs for nearby, but more vulnerable, localities. Repeated exposure to sublethal heat stress can increase coral resilience—a phenomenon referred to as environmental memory[9,90,91]. This suggests that under certain conditions corals might develop greater tolerance to future thermal stress. However, while our understanding of this mechanism and its trade-offs with other ecological processes is still limited, exposure to DHW values above the critical thresholds (for example, bleaching alert level II) continues to be a major concern, often followed by high coral mortality, especially in the absence of sufficient recovery periods between stress events[4].

Deciding where to restore is a fundamental planning detail, yet it is only one facet of the process. Of equal importance is careful selection of the reared and/or transplanted coral species. In principle, the choice of candidate species should be based on sound ecological reasoning and consider important aspects for survival such as susceptibility to predation, disease and rising temperatures[92]. However, in most cases the target corals have limited taxonomic and functional diversity and their selection is not supported by ecological reasoning—their role in ecosystem processes, resilience to environmental stressors and capacity to support biodiversity and promote long-term reef recovery. The main goal of most restoration projects is to re-establish coral cover, with less attention devoted to maximizing biodiversity and ecological function[60]. This limitation is compounded by the challenges of rearing and outplanting some coral species; hence, fast-growing genera such as *Acropora* are preferred[93].

The coral taxa used in all restoration projects[21] accounted for only 0.04% of the local coral diversity on average. Even for successful restoration actions, it is unlikely that such limited taxonomic diversity could provide the range of ecological functions needed to maximize community resilience[92,94]. Different corals support many specialized interactions (such as those with the Chaetodontidae and coral-dwelling Gobiidae[95,96] or with invertebrate epifauna such as highly coral-specific *Tetralia* species crabs)[97] that could disappear following declining coral diversity regardless of total cover. At the global scale, coral diversity drives fish diversity, with the expectation of a proportional loss of fish diversity with the loss of coral diversity, corroborating the idea that all coral species are not ecologically equivalent[6,72].

The planning, evaluation, cost and scalability challenges (Fig. 1) raise important questions about the current effectiveness of coral restoration as an effective response to ongoing coral decline[98]. Our analyses suggest that although coral restoration has the potential to be a valuable tool in certain circumstances it is not yet feasible to scale it up sufficiently to have meaningful, long-term and positive effects on coral reef ecosystems. This reality check should stimulate constructive debate about maximizing the utility of restoration, particularly in combination with broader strategies to address climate change and other threats.

Considering the constraints facing current restoration methods, our findings underscore the importance of exploring indirect interventions to support coral resilience while addressing $CO_2$ reduction on a broader scale. For instance, combined approaches targeting both sea- and land-based stressors[5] such as marine protected areas[99] and water quality remediation[100,101] could create synergistic benefits for reefs while also supporting local human communities with incentives for

conservation. Although beyond the immediate scope of this paper, reinforcing complementary strategies could bolster ecosystem resilience, extending the reach of coral restoration efforts.

## Methods

### Data

We obtained data for the global distribution of coral reefs from the UN Environment Programme World Conservation Monitoring Centre[102] at a resolution of 0.5° latitude × 0.5° longitude. We integrated data for coral restoration events from Boström-Einarsson et al.[21]—the most comprehensive database of coral restoration available (Extended Data Fig. 1). This dataset includes the location and timing of each restoration action, the broad category of the disturbance that caused coral mortality and generated the need for restoration, the species or genera used, the restoration technique (which we reclassified into four categories: coral gardening using nurseries, direct transplantation, building of artificial reefs and larval enhancement), the monitoring duration of restored colonies and coral survival in the monitored colonies.

We then obtained data on different potential sources of detrimental impacts on coral communities. We hypothesized that these variables affect site selection and restoration success. For example, highly impacted sites might be more degraded, thereby warranting restoration[69]. Additionally, sites more vulnerable to impacts face higher risks of conservation interventions being jeopardized by external stressors.

We accounted for thermal anomalies recorded before, during and after the restoration. We retrieved thermal anomalies from 1986–2021 from NOAA. We focused on the bleaching alert levels, identifying for each reef locality the number of recorded events of severe bleaching alert (levels I and II) per year. Specifically, bleaching alert level I identifies localities experiencing instantaneous (that is, daily) heat stress > 1 °C and DHW = 4–8 °C-weeks, whereas bleaching alert level II indicates heat stress > 1 °C and DHW ≥ 8 °C-weeks. We obtained data describing anthropogenic pressures from Halpern et al.[71]. This dataset includes global and local threats to marine ecosystems. We computed both means and trends of cumulative impacts from 2003–2013 for each reef location (see ref. 71).

We integrated two measures of the accessibility of each locality to humans: remoteness (that is, the shortest travel distance from the closest large human settlement (a locality with a population density of ≥1,500 inhabitants per km² or a built-up density of >50% and ≥50,000 inhabitants))[72] and gravity (quantifying human accessibility to reefs as a function of distance (travel time) and population size)[73]. These two measures offer complementary information. Remoteness is a standardized proxy for the technical and logistical challenges associated with implementing restoration, whereas gravity is an additional proxy for human disturbance that quantifies the magnitude of potential human–reef interactions. Proximity to large human settlements can have the opposite effects of hindering restoration success through increased disturbances and simplifying operations for the implementation and maintenance of restoration. We also included information on coral diversity (quantified as species richness) for each locality using the coral species range provided by the International Union for Conservation of Nature (www.iucnredlist.org).

Before each of the modelling approaches described in the following sections, we explored pairwise collinearity among all of the variables to exclude potentially redundant predictors. The correlations ($R^2$) were <0.7 for all pairs (see Supplementary Figs. 1 and 2), so we eventually opted to include all of the variables in the models.

### Choice of restoration sites

We searched for potential determinants driving the selection of the restoration localities using a machine-learning approach—boosted regression trees[103]—to generate models differentiating restored sites from other reef locations based on a combination of hypothesized predictors (the response is therefore a binary variable indicating whether the target site had been restored or not). We hypothesized that sites more impacted by anthropogenic and climatic stressors (that is, commercial demersal destructive fishing, commercial demersal non-destructive high- and low-bycatch fishing, pelagic high- and low-bycatch fishing, artisanal fishing, sea surface temperature increase, ocean acidification, sea level rise, shipping, nutrient pollution, chemical pollution, direct human damage and light) could be more targeted by conservation actions. We also expected that accessibility to the sites (remoteness and gravity) could be important because they measure the practical challenges of implementing restoration (for example, transporting materials, ensuring continuous nursery monitoring and so on), as well as reef exposure to human impacts. We also included local coral diversity under the hypothesis that species-rich sites might be more likely to be flagged as conservation priorities.

We used cross-validation to calibrate the model and assess its accuracy. First, we identified the best model parameterization by exploring how model accuracy varied for different combinations of learning rate (0.01, 0.001 or 0.0001), bag fraction (0.5, 0.7 or 0.8) and tree complexity (1, 2, 3, 4 or 5). We generated ten models for each combination of parameters. For each model–parameter combination, we also explored the effect of the number of trees on model performance by using the function gbm.step from the dismo package[104], training models by varying tree number from 50–10,000 with incremental steps of ten trees and finally retaining the one model minimizing holdout deviance.

We trained the models on a set of observations including 80% of randomly selected presences (restored sites) and 80% of randomly selected absences (non-restored sites) and tested model performance on the remaining observations. To address the class imbalance in the training data, where restored sites were less common than non-restored sites, we applied site weights inversely proportional to the prevalence of each class. This approach ensured that the less common restored sites contributed more to the model training, whereas non-restored sites were downweighted to prevent the model from becoming biased towards the majority class. To account for potential issues arising from spatial autocorrelation in the probability of a site being selected for restoration, instead of using the full set of reef locations, we generated a random set of locations characterized by spatial independence in their restored or non-restored state for each model. For this, we started from a randomly selected reef locality and then iterated through all of the other reef localities one at a time, adding a locality to the set only if it was >150 km (haversine distance) from the closest locality already in the set. We then tested spatial independence between the localities included in the set using join count statistics[105] to assess the probability that any two neighbouring sites had the same status (restored or non-restored) consistent with random expectation (the expected probability if the observed states were randomly reallocated across localities). Here we used the function joincount.multi from the spdep package[106]. We discarded the sets not satisfying this condition, replicating the random site selection procedure until it generated a valid set.

We evaluated model accuracy by applying each model to the testing set, comparing observed and predicted classification values and computing true skills statistics (TSSs)[107]. We explored probability thresholds for class assignment in 0.001, 0.002, …, 0.900 and then selected the threshold maximizing the TSS. We computed the mean of the TSS scores across replicates of parameter combinations and then selected the parameterization giving the highest TSS (learning rate = 0.001, bag fraction = 0.5 and tree complexity = 3). We then trained and tested 1,000 models with the same procedure as above and the optimal parameterization. For each model, we computed (cross-validated) type I and II (false positive and negative) error rates and the TSS. We also replicated this step by generating training sets including 50% of observations instead of 80%. The substantial reduction in the amount of training information provided to the model resulted in a small reduction in the model accuracy, yielding an average ± s.d. TSS of 0.40 ± 0.12.

We derived a complete model (with best parameterization) on the full dataset, which we used to explore variable importance. We computed the relative influence of each independent variable and used partial dependency plots[108] generated with the R package pdp[109] to show individual relationships between the independent variables and the predicted probability of site selection.

#### Determinants of short-term restoration success

We combined information on the duration of post-restoration monitoring and the relative survival of the monitored colonies to obtain a standardized proxy for restoration success (Extended Data Fig. 4). Based on empirical studies, we assumed an exponential decay relationship between coral mortality and coral size, with mortality high in the early phases of growth and decreasing as corals grow. We calibrated the relationship based on an empirical curve of percentage coral survival versus time from the literature (Supplementary Fig. 1). We then used the curve to quantify the percentage of surviving coral colonies (starting from a class size of 5–10 cm, which is the typical size used in coral restoration) after $n$ post-monitoring months. We then derived a proxy of restoration success ($S_r$) by quantifying the deviation of the observed survival ($S_o$) from the expected survival ($S_e$) as:

$$S_r = \log_e\left(1 + 100 - \frac{(S_e - S_o)}{S_e}100\right)$$

(Supplementary Fig. 1). We then used this measure as a dependent variable in the boosted regression trees testing the determinants of restoration success. As candidate independent variables, we included the various measures of means and trends in human impacts (12 variables), the number of pre- and post-restoration severe bleaching alerts recorded in the target locations (within windows of 5 years before and after the restoration date), local coral diversity, remoteness, gravity and coral restoration technique (four binary variables indicating the use of coral gardening, transplantation, artificial reef building and larval enhancement). Because we detected weak evidence of spatial autocorrelation in the measure of restoration success (Moran's $IP$ = 0.049), we reduced the set of observations used to train and test the models by repeating the procedure described in the previous section ('Choice of restoration sites'). Considering the weak evidence for spatial autocorrelation detected in the full dataset (and its smaller size), we used a distance threshold of 1 km (that is, we excluded only overlapping sites from the subset). Considering that the target variable was continuous (that is, restoration success), we tested each resulting subset for spatial autocorrelation using Moran's $IP$ (retaining sets with $P$ > 0.05).

Before assessing model performance through cross-validation (including 80% of observations in the training set and the remaining 20% in the testing set), we first tuned the model by exploring different parameterizations by varying the learning rate, bag fraction, tree complexity and number of trees using the same procedure and parameter value combinations we used for the restoration site selection model. We evaluated model accuracy in terms of the goodness of fit (Pearson's $R^2$) of predicted versus observed success in each testing set and then selected the parameterization leading to the highest mean accuracy (learning rate = 0.001, bag fraction = 0.7 and tree complexity = 5). We then used this parameterization to train and test 1,000 models. Because the average accuracy of these models gave a low ($R^2$ < 0.05) goodness of fit, we did not explore variable importance as possibly misleading.

#### Exploring the future fate of restored reefs

We obtained cumulative heat stress data from ref. 78 in the form of DHW values calculated over a rolling 12-week (84-day) window. These data are provided at the same spatial resolution as our dataset (0.5° latitude × 0.5° longitude), hence requiring no adjustments. We focused on an intermediate climate projection (SSP2-4.5; middle of the road) and used the model ensemble product (combining five different models)

provided by the authors. For each locality, we counted the number of years from 2015–2100 in which the maximum yearly DHW value was ≥20 °C-weeks. At this threshold, mass coral mortality (>80% of corals) is expected[4]. We also took note of the first year in which such thresholds occurred. We then compared projected heat stress data between restored and non-restored localities.

#### Reporting summary

Further information on research design is available in the Nature Portfolio Reporting Summary linked to this article.

### Data availability

All of the data and code needed to replicate the analyses are available at https://doi.org/10.5281/zenodo.14760258 (ref. 110). Source data are provided with this paper.

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

## Acknowledgements

We are indebted to T. Hughes for providing several valuable ideas and suggestions in relation to an earlier version of this manuscript.

## Author contributions

G.S., C.M. and C.J.A.B. designed the research and performed the analyses. C.M. and G.S. collected the data and wrote the first draft of this manuscript. G.S. produced the figures. C.J.A.B., S.M., F.M. and M.C. contributed to the writing, reviewed the analyses, interpreted the results and provided conceptual input. All authors approved the final version of the manuscript.

# Article

## Competing interests

The authors declare no competing interests.

## Additional information

**Extended data** is available for this paper at

**Supplementary information** The online version contains supplementary
material available at https://doi.org/10.1038/s41559-025-02667-x.

**Correspondence and requests for materials** should be addressed to
Giovanni Strona.

**Peer review information** *Nature Ecology & Evolution* thanks Emma Camp,
Gareth Williams and the other, anonymous, reviewer(s) for their contribution
to the peer review of this work. Peer reviewer reports are available.

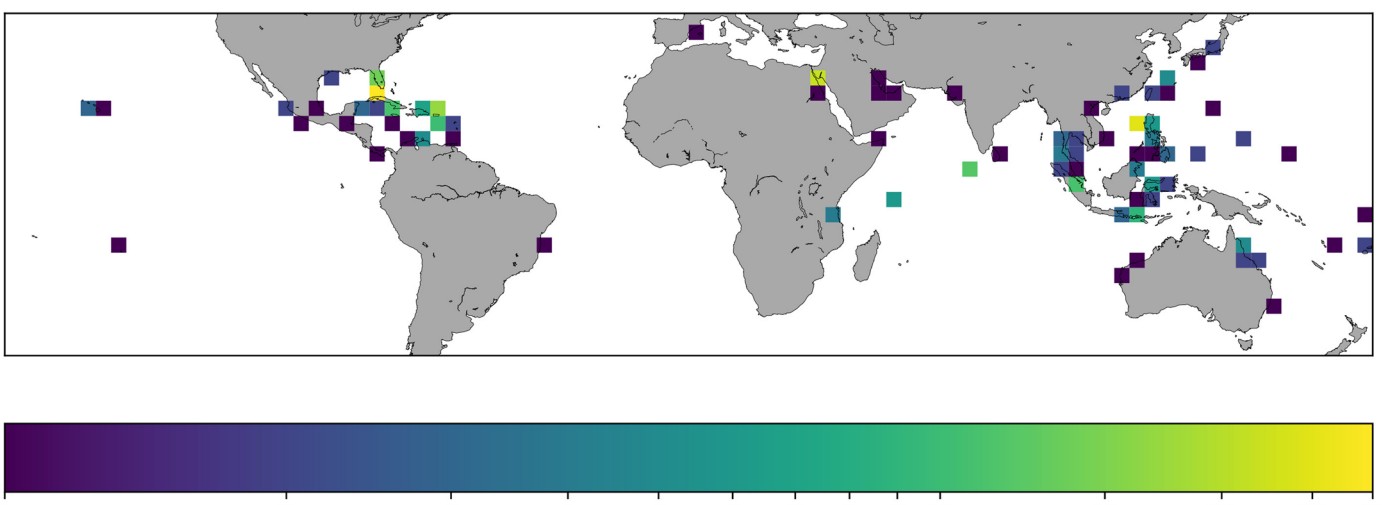

**Extended Data Fig. 1 | Global distribution of restored sites from the dataset.** We aggregated data from Boström-Einarsson et al.[21] at a resolution of 4° × 4° latitude/longitude to ease visualization.

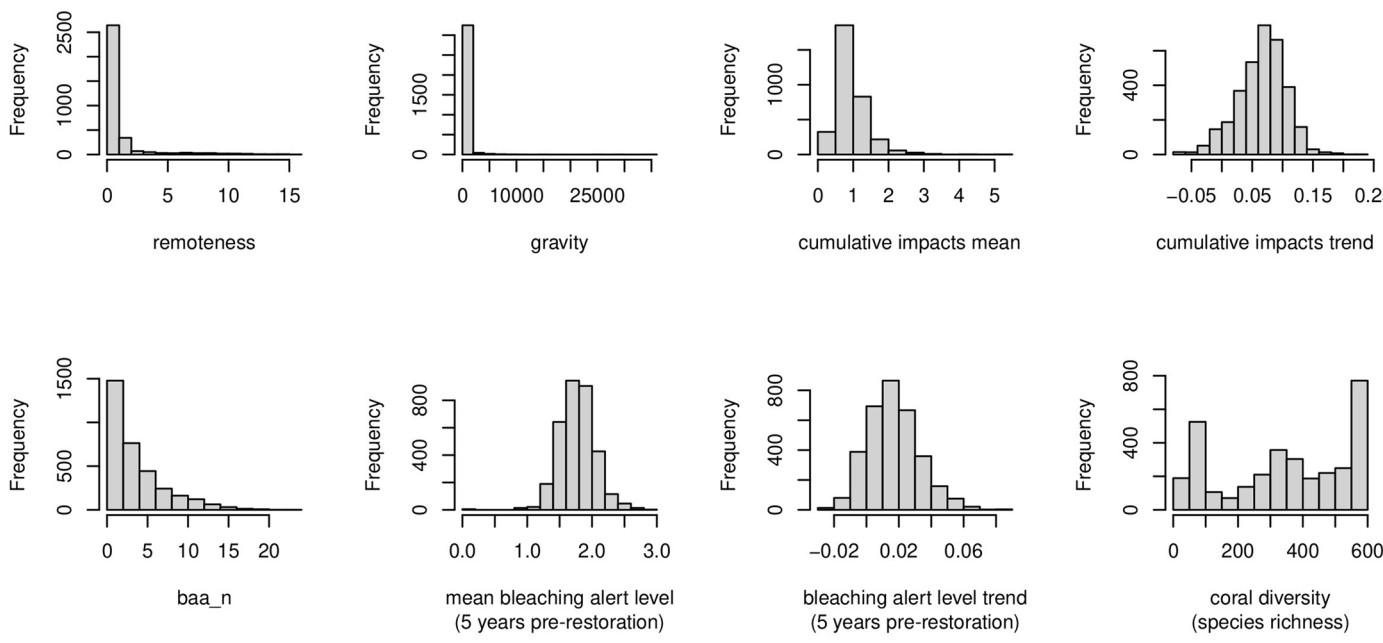

**Extended Data Fig. 2 | Frequency distribution of the independent variables used to predict the choice of restoration sites.** Total *n* observations = 3324.

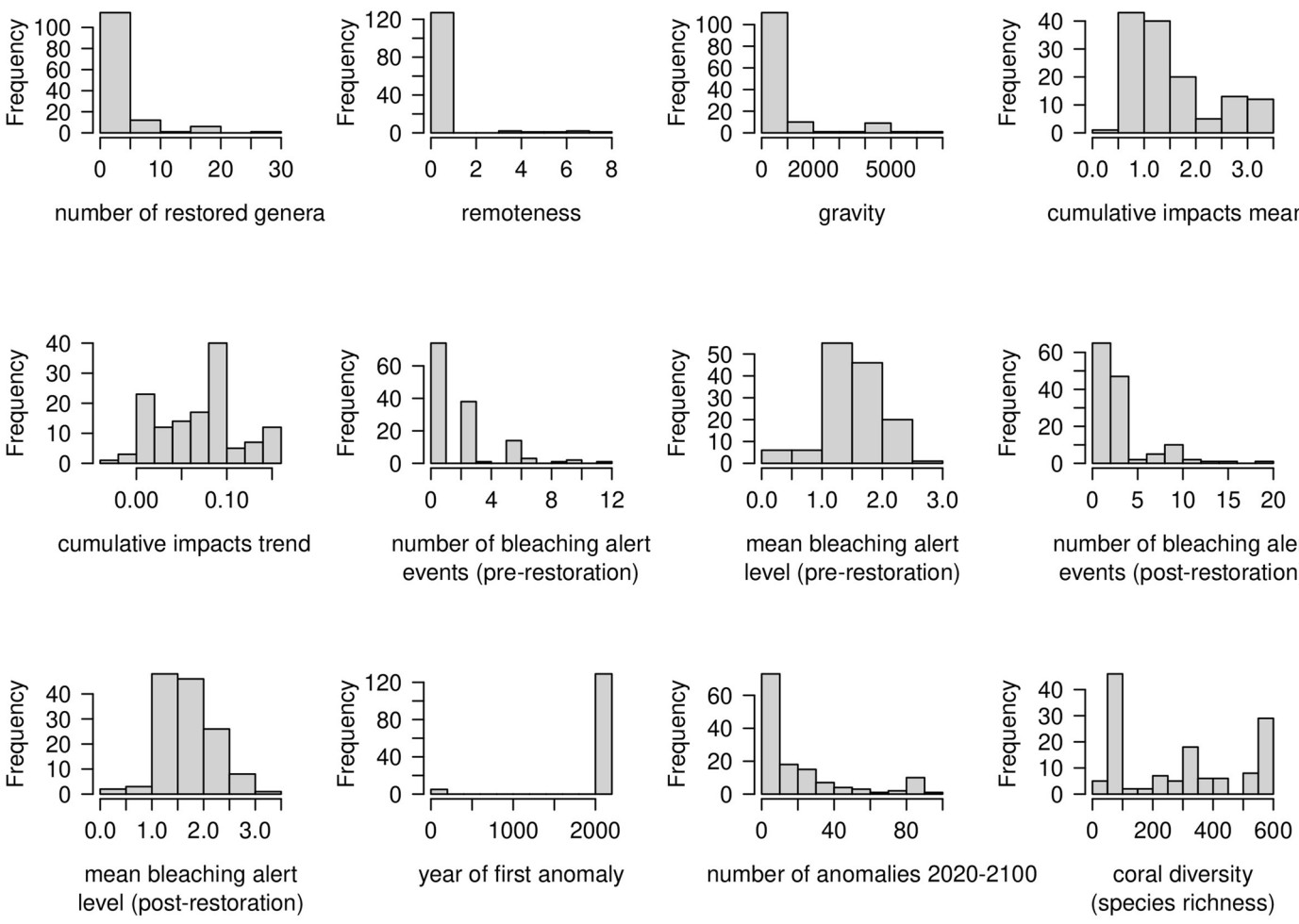

**Extended Data Fig. 3 | Frequency distribution of the independent variables used to predict coral restoration success.** Total *n* observations = 134.

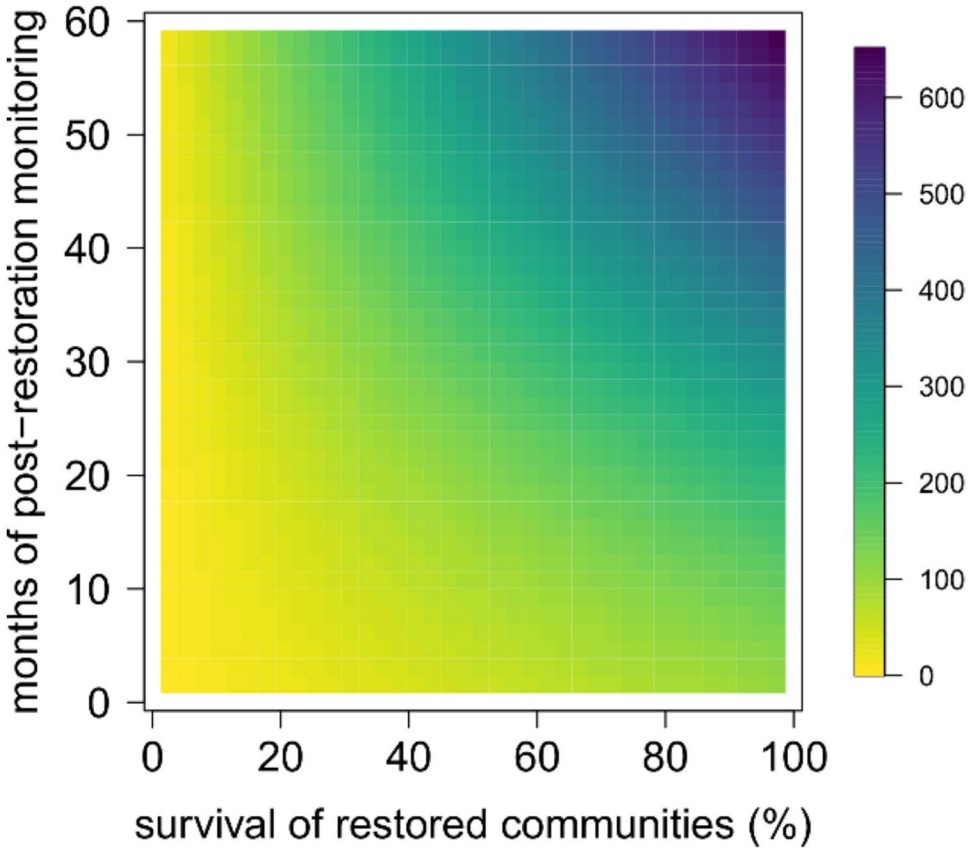

**Extended Data Fig. 4 | Measure of standardized success in coral restoration actions ($S_r$).** $S_r$ is computed as the deviation of the observed survival ($S_o$) from expected survival ($S_e$) (after a given number of post restoration months) as $S_r = \log_e(1 + 100 - \frac{(S_e - S_o)}{S_e} 100)$. $S_e$ is computed based on an empirical curve of coral survival % *versus* time obtained from literature.

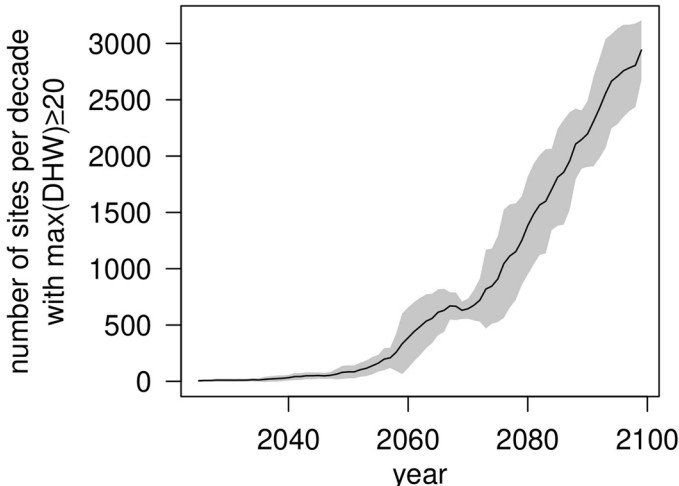

**Extended Data Fig. 5 | Mean number of sites experiencing a maximum yearly degree heating weeks ≥ 20 per decade.** The plot shows the mean number of sites with predicted maximum yearly DHW ≥ 20 globally within a moving time window of 10 years. We mapped reef localities at a resolution of 0.5° × 0.5° latitude/longitude (*n* = 3780). The moving time window started at year 2015, so the first reported year is 2025. Predictions are based on an intermediate CMIP6 climate projection (SSP2-4.5, 'middle of the road'; see Methods for details), with DHW obtained from ref. 79.

**Extended Data Table 1 | Cost estimates to rehabilitate 10% of coral reefs degraded between 2009 and 2018**

| restoration technique | n | estimates of restoration cost (US$/ha) | | | | estimated cost to restore 1,170 km² of reef (billion US$) | | |
|---|---|---|---|---|---|---|---|---|
| | | median | SD | min | max | median | min | max |
| direct transplantation | 20 | 218,305 | 2,339,609 | 9,198 | 8,382,653 | 25.5 | 1.1 | 980.8 |
| larval enhancement | 6 | 523,162 | 1,878,894 | 6,262 | 4,333,826 | 61.2 | 0.7 | 507.1 |
| coral gardening (overall) | 3 | 351,661 | 136,601 | 130,000 | 379,139 | 41.1 | 15.2 | 44.4 |
| substrate addition artificial reef | 10 | 3,341,754 | 44,100,144 | 14,076 | 142,667,803 | 391.0 | 1.6 | 16692.1 |
| substrate stabilization | 8 | 370,986 | 9,040,923 | 91,044 | 26,100,000 | 43.4 | 10.7 | 3053.7 |
| coral gardening (collection and nursery) | 5 | 28,075 | 20,472 | 9,262 | 56,150 | 3.3 | 1.1 | 6.6 |
| coral gardening (transplantation) | 2 | 761,864 | 10,33,831 | 30,835 | 14,92,893 | 89.1 | 3.6 | 174.7 |

Data based on Table 1 from Bayraktarov et al.[40]

# Reporting Summary

## Statistics

For all statistical analyses, confirm that the following items are present in the figure legend, table legend, main text, or Methods section.

| n/a | Confirmed | |
|---|---|---|
| ☐ | ☒ | The exact sample size (*n*) for each experimental group/condition, given as a discrete number and unit of measurement |
| ☐ | ☒ | A statement on whether measurements were taken from distinct samples or whether the same sample was measured repeatedly |
| ☐ | ☒ | The statistical test(s) used AND whether they are one- or two-sided *Only common tests should be described solely by name; describe more complex techniques in the Methods section.* |
| ☐ | ☒ | A description of all covariates tested |
| ☐ | ☒ | A description of any assumptions or corrections, such as tests of normality and adjustment for multiple comparisons |
| ☐ | ☒ | A full description of the statistical parameters including central tendency (e.g. means) or other basic estimates (e.g. regression coefficient) AND variation (e.g. standard deviation) or associated estimates of uncertainty (e.g. confidence intervals) |
| ☐ | ☒ | For null hypothesis testing, the test statistic (e.g. *F*, *t*, *r*) with confidence intervals, effect sizes, degrees of freedom and *P* value noted *Give P values as exact values whenever suitable.* |
| ☒ | ☐ | For Bayesian analysis, information on the choice of priors and Markov chain Monte Carlo settings |
| ☒ | ☐ | For hierarchical and complex designs, identification of the appropriate level for tests and full reporting of outcomes |
| ☐ | ☒ | Estimates of effect sizes (e.g. Cohen's *d*, Pearson's *r*), indicating how they were calculated |

*Our web collection on statistics for biologists contains articles on many of the points above.*

## Software and code

Policy information about availability of computer code

| Data collection | Data were downloaded from online sources with the aid of no software. |
|---|---|
| Data analysis | Data analyses were conducted in Python and R. We provide the all the data and code needed to replicate the analyses at github.com/giovannistrona/coral_restoration |

For manuscripts utilizing custom algorithms or software that are central to the research but not yet described in published literature, software must be made available to editors and reviewers. We strongly encourage code deposition in a community repository (e.g. GitHub). See the Nature Portfolio guidelines for submitting code & software for further information.

## Data

Policy information about availability of data

All manuscripts must include a data availability statement. This statement should provide the following information, where applicable:
- Accession codes, unique identifiers, or web links for publicly available datasets
- A description of any restrictions on data availability
- For clinical datasets or third party data, please ensure that the statement adheres to our policy

We obtained data for the global distribution of coral reefs from UNEP-WCMC (https://data-gis.unep-wcmc.org/portal/home/item.html?id=9f6664a6720f420580f5f54f7925dfee)
We obtained data for coral restoration events from Boström-Einarsson et al. 2020 (https://datadryad.org/stash/dataset/doi:10.5061/dryad.p6r3816)

We retrieved data on thermal anomalies and bleaching alert levels from NOOA (https://www.star.nesdis.noaa.gov/pub/sod/mecb/crw/data/5km/v3.1_op/nc/v1.0/annual)
We retrieved ocean impacts from Halpern et al. 2019 (https://knb.ecoinformatics.org/view/doi:10.5063/F12B8WBS)
We retrieved the map of gravity from Cinner et al. 2018 (https://researchdata.jcu.edu.au/published/a9167f52dba39f693f55ae68a0a5dccf/)
We obtained future DHW data from Mellin et al. 2024 (https://adelaide.figshare.com/articles/dataset/Global_projections_of_sea_surface_temperature_and_coral_bleaching_risk_in_the_21st_century/25143128)

# Research involving human participants, their data, or biological material

Policy information about studies with human participants or human data. See also policy information about sex, gender (identity/presentation), and sexual orientation and race, ethnicity and racism.

**Reporting on sex and gender**
*Use the terms sex (biological attribute) and gender (shaped by social and cultural circumstances) carefully in order to avoid confusing both terms. Indicate if findings apply to only one sex or gender; describe whether sex and gender were considered in study design; whether sex and/or gender was determined based on self-reporting or assigned and methods used.*
*Provide in the source data disaggregated sex and gender data, where this information has been collected, and if consent has been obtained for sharing of individual-level data; provide overall numbers in this Reporting Summary. Please state if this information has not been collected.*
*Report sex- and gender-based analyses where performed, justify reasons for lack of sex- and gender-based analysis.*

**Reporting on race, ethnicity, or other socially relevant groupings**
*Please specify the socially constructed or socially relevant categorization variable(s) used in your manuscript and explain why they were used. Please note that such variables should not be used as proxies for other socially constructed/relevant variables (for example, race or ethnicity should not be used as a proxy for socioeconomic status).*
*Provide clear definitions of the relevant terms used, how they were provided (by the participants/respondents, the researchers, or third parties), and the method(s) used to classify people into the different categories (e.g. self-report, census or administrative data, social media data, etc.)*
*Please provide details about how you controlled for confounding variables in your analyses.*

**Population characteristics**
*Describe the covariate-relevant population characteristics of the human research participants (e.g. age, genotypic information, past and current diagnosis and treatment categories). If you filled out the behavioural & social sciences study design questions and have nothing to add here, write "See above."*

**Recruitment**
*Describe how participants were recruited. Outline any potential self-selection bias or other biases that may be present and how these are likely to impact results.*

**Ethics oversight**
*Identify the organization(s) that approved the study protocol.*

Note that full information on the approval of the study protocol must also be provided in the manuscript.

# Field-specific reporting

Please select the one below that is the best fit for your research. If you are not sure, read the appropriate sections before making your selection.

☐ Life sciences    ☐ Behavioural & social sciences    ☒ Ecological, evolutionary & environmental sciences

For a reference copy of the document with all sections, see nature.com/documents/nr-reporting-summary-flat.pdf

# Ecological, evolutionary & environmental sciences study design

All studies must disclose on these points even when the disclosure is negative.

**Study description** — We combined various datasets to explore the determinants of coral restoration success, the motivations behind the choice of coral reef sites to be restored, and the potential fate of restored reef under future scenarios of climate change.

**Research sample** — We combined spatially explicit environmental and climatic information on all reef sites globally at a resolution of 0.5 x 0.5 latitude/longitude degrees with point data providing various information for 220 coral restoration projects.

**Sampling strategy** — We included all available data from the dataset by Boström-Einarsson et al. (2020)

**Data collection** — Data were collected from online (open) repositories (as described in the data availability section), by GS

**Timing and spatial scale** — The study includes historical data from 1979 to 2018 and future projections until 2100

**Data exclusions** — We excluded no data

**Reproducibility** — The analyses are fully reproducible (with code and data openly available at  github.com/giovannistrona/coral_restoration)

**Randomization** — We performed cross validation to evaluate the accuracy of boosted regression tree models, by randomly allocating 80% of available

| Randomization | data in the training set and the remaining 20% in the testing set. We also performed a random spatial resampling of coral restoration localities to cope with spatial autocorrelation in the evaluation of models' accuracy. In all cases, records and localities were assigned to random samples with uniform/equal probability. |
|---|---|
| Blinding | No statistical analysis or data collection procedure required blinding. |

Did the study involve field work? ☐ Yes ☒ No

# Reporting for specific materials, systems and methods

We require information from authors about some types of materials, experimental systems and methods used in many studies. Here, indicate whether each material, system or method listed is relevant to your study. If you are not sure if a list item applies to your research, read the appropriate section before selecting a response.

## Materials & experimental systems

| n/a | Involved in the study |
|---|---|
| ☒ ☐ | Antibodies |
| ☒ ☐ | Eukaryotic cell lines |
| ☒ ☐ | Palaeontology and archaeology |
| ☒ ☐ | Animals and other organisms |
| ☒ ☐ | Clinical data |
| ☒ ☐ | Dual use research of concern |
| ☒ ☐ | Plants |

## Methods

| n/a | Involved in the study |
|---|---|
| ☒ ☐ | ChIP-seq |
| ☒ ☐ | Flow cytometry |
| ☒ ☐ | MRI-based neuroimaging |

## Plants

| Seed stocks | *Report on the source of all seed stocks or other plant material used. If applicable, state the seed stock centre and catalogue number. If plant specimens were collected from the field, describe the collection location, date and sampling procedures.* |
|---|---|
| Novel plant genotypes | *Describe the methods by which all novel plant genotypes were produced. This includes those generated by transgenic approaches, gene editing, chemical/radiation-based mutagenesis and hybridization. For transgenic lines, describe the transformation method, the number of independent lines analyzed and the generation upon which experiments were performed. For gene-edited lines, describe the editor used, the endogenous sequence targeted for editing, the targeting guide RNA sequence (if applicable) and how the editor was applied.* |
| Authentication | *Describe any authentication procedures for each seed stock used or novel genotype generated. Describe any experiments used to assess the effect of a mutation and, where applicable, how potential secondary effects (e.g. second site T-DNA insertions, mosaicism, off-target gene editing) were examined.* |

