## [Peer Review File · Nature Ecology & Evolution]

Restoration cannot be scaled up globally to save reefs from loss and degradation

Corresponding Author: Dr Giovanni Strona

Version 0:

Decision Letter:

20th August 2024

Dear Dr Strona,

Your manuscript entitled "Restoration alone will not save coral reefs from climate change" has now been seen by three reviewers, whose comments are attached. The reviewers have raised a number of concerns which will need to be addressed before we can offer publication in Nature Ecology & Evolution. We will therefore need to see your responses to these concerns, along with a revised manuscript, before we can reach a final decision regarding publication.

We therefore invite you to revise your manuscript taking into account all reviewer and editor comments. Please highlight all changes in the manuscript text file.

* If you have not done so already please begin to revise your manuscript so that it conforms to our Article format instructions at <http://www.nature.com/natecolevol/info/final-submission>. Refer also to any guidelines provided in this letter.

Link Redacted

Nature Ecology & Evolution is committed to improving transparency in authorship. As part of our efforts in this direction, we are now requesting that all authors identified as 'corresponding author' on published papers create and link their Open Researcher and Contributor Identifier (ORCID) with their account on the Manuscript Tracking System (MTS), prior to acceptance. ORCID helps the scientific community achieve unambiguous attribution of all scholarly contributions. You can create and link your ORCID from the home page of the MTS by clicking on 'Modify my Springer Nature account'. For more information please visit www.springernature.com/orcid.

[redacted]

Reviewer comments:

Reviewer #1 (Remarks to the Author):

Mula et al. assess the efficacy of coral reef restoration using a globally extensive data set. Given the very significant degradation of coral reefs world-wide, the ongoing and increasing risks to coral reefs from a variety of stressors, and particularly, the very high cost of restoration, their paper is profoundly important and should be carefully considered by practitioners in this field. Generally, their conclusions align with much of what many of us have been thinking for quite some time but have largely remained silent. Well done, Mula et al! This paper also provides an example of how to approach the study and prioritization of restoration more broadly in other ecosystems. As such, this paper will be of considerable interest to scientists and other practitioners far beyond just coral reefs. This broader applicability could be emphasised more strongly.

I offer a few additional comments below that I think could strengthen an already excellent paper. They should not be interpreted in any way that diminishes my strong recommendation to publish this paper.

I enjoyed the economic analysis of the costs of restoration at scale. This alone should cause anyone advocating restoration through direct intervention to carefully consider other options. I think it would be very helpful if the authors provided some additional information as to what is included by their referenced sources. For example, are the costs of assisted evolution (e.g., genetic engineering) and assisted gene flow included. Both are inordinately expensive, are nearly impossible to scale geographically, are applicable to a vanishingly small number of species, and nearly always do not properly consider the evolutionary dynamics that can impose considerable maladaptation in their application.

I also would like to see the authors be a bit braver and conclude their paper with a set of next steps/best options for research and practice in this area for both coral reefs and beyond. For example, it is clear and I agree, they do not believe that direct intervention is likely to succeed without considerable modification to the methods being currently applied. Therefore, what about indirect intervention before CO2 can be adequately drawn down such as say combinations of MPAs and water quality remediation that also benefits landholders. While beyond the scope of this paper to consider in depth, it might also be useful to include in this discussion, indirect interventions that will also fail such as the direct removal of Crown-of-Thorns starfish at scale.

Julian Caley

Reviewer #2 (Remarks to the Author):

The paper by Mula et al. models' locations where coral restoration has occurred from a 2020 dataset and then considers environmental impacts predicted to occur relative to these sites. In principle the activity they have undertaken is insightful and could provide useful information, e.g. mapping vulnerable areas to future change relative to restoration hotspots. I wish to note, as a coral scientist involved in restoration, their title, "restoration alone will not save coral reefs from climate change" is a statement I completely agree with.

However, I have serious concerns on how the paper has been framed, which fails to capture the reality of why restoration is occurring where it is, e.g., the societal element of coral reefs. Further, there are broad statements about restoration that cannot be generalised. For example, "Coral restoration is gaining popularity as a mitigation strategy". What it is apparently mitigating against was not specified, but time and time again in the coral literature, restoration is not described as a substitution for climate action. Restoration exists as a continuum of approaches from mitigation of a stress through to rehabilitation (see Suggett et al., 2024). This nuanced understanding of the intent of restoration is crucial, as the current framing of the paper reads as through it is trying to disprove restorations validity to mitigate climate change, which is not a universal narrative being put forward by the restoration community. Without a more nuanced discussion of restoration intent, I cannot recommend the work for publication. I have several additional comments below:

Line 25: "possible mitigation strategy" – mitigation to what? If climate action, please remove as this is not a common intent of restoration and fails to capture the continuum along which restoration exists (see Suggett et al. 2024).

Line 26: Clarify what the mass-mortality refers to.

Line 27: What is meant by cost? Societal, financial, ecological? Please clarify.

Line 27: "The costs of restoration are as massive as yearly costs" – This cannot be generalised for all reefs and all restoration practices. These general statements should be removed or rephrased with specific, evidenced based examples.

Line 28: Is global planning even appropriate? Who would make these choices, and what about the social licence of local communities and their voice in decision making?

Line 31: "We highlight how such a plan" – What is the plan? It is unclear.

Line 34: Yes, but what about the flipside of the argument, that close to human settlement is where you have the highest social needs, and therefore desire and need to restore reefs. This is a crucial viewpoint that has been missed from the paper.

Line 35: “Influenced by relevant ecological and environmental predictors” – can an example be provided to provide clarity.

Line 38: “to reinforce joint development of restoration guidelines that go beyond local objectives” Is this appropriate? What about the voice of local people in decision making and the local site-specific context that shapes why restoration is being undertaken, and therefore what success is.

Line 44: Update reference 2; it is 1 Billion people (A. Sing Wong, S. Vrontos, M. L. Taylor, An assessment of people living by coral reefs over space and time. *Glob. Chang. Biol.* 28, 7139–7153 (2022).)

Line 46: Needs a reference.

Line 51-52: As per my comment in the abstract, this is not a fair representation of the continuum of approaches that restoration incorporates. Please balance this narrative to reflect this. Many restoration studies explicitly state that restoration is not a replacement for climate action, for example in Suggett et al. (2024), “Ultimately, there is no “one size fits all” method for restoring coral reefs or measuring success, nor will we restore our way out of the climate crisis”. Such balance should be provided in the paper.

Line 60: There is a lot of literature since the 2020 Bostrom-Einarsson et al. work discussing the current state of reef restoration which is relevant here. This is summarised again in the Suggett et al. (2024), where they highlight that the life stage of restoration (eg science and R and D) relative to industrial application at scale are being incorrectly compared to then decide restoration has failed. To work out the best approach, fast-fail science is being applied, and this is a necessary path if any scalable solutions are to be identified. Acknowledgement of this point should be included.

Line 64: Could also include examples of restoration where reef function has been returned, for example calcification: Nuñez Lendo, C.I., Suggett, D.J., Boote, C., McArdle, A., Nicholson, F., Fisher, E.E., Smith, D. and Camp, E.F., 2024. Carbonate budgets induced by coral restoration of a Great Barrier Reef site following cyclone damage. *Frontiers in Marine Science*, 10, p.1298411.

Lange, I.D., Razak, T.B., Perry, C.T., Maulana, P.B., Prasetya, M.E. and Lamont, T.A., 2024. Coral restoration can drive rapid reef carbonate budget recovery. *Current Biology*, 34(6), pp.1341-1348.

Line 73: Other examples of restoration with much lower costs not included, eg Scott et al. 2024; Scott, R.I., Edmondson, J., Camp, E.F., Agius, T., Coulthard, P., Edmondson, J., Edmondson, K., Hosp, R., Howlett, L., Roper, C.D. and Suggett, D.J., 2024. Cost-effectiveness of tourism-led coral planting at scale on the northern Great Barrier Reef. *Restoration Ecology*, 32(4), p.e14137.

Line 73: It would be balanced to also consider the cost of no action to local communities that rely on healthy, functioning coral reefs.

Figure 1. It would be balanced to show what the loss of value would be if a reef is lost. See the example considered of value returned relative to restoration undertaken in Suggett et al. 2023

Suggett, D.J., Edwards, M., Cotton, D., Hein, M. and Camp, E.F., 2023. An integrative framework for sustainable coral reef restoration. *One Earth*, 6(6), pp.666-681.

Line 95: Provide examples of where restoration has been successful locally.

Line 96: Same comment about mitigation being used here inappropriately.

Lines 100-106: These challenges have been highlighted in prior studies that should be acknowledge, e.g. the challenges being put forward are not newly described challenges.

Line 125: What is the definition of success here? And how was this standardised for the 220 studies investigated. If one program had success as upskilling local communities and another was to increase biodiversity, how were these differences considered?

Line 133: Is the data available?

Line 134: “We hypothesise that some of these variables” – Please specify which ones.

Line 141-145: But what about the societal aspect of reefs. This is a missing component of the narrative you have for restoration site selection, yet it is key as reefs are socioecological systems.

Line 147: What are the 14 stressors?

Line 177: Without defining success and standardising this across the 220 studies, how can the drivers of restoration success be determined?

Line 178: What are the 9 indicators?

Line 187: Please clarify with higher or lower coral diversity.

Line 200-209: Is there information on mortality. What about narrative on environmental memory and the fact that some heat exposure can increase tolerance to future stress. While I agree exposure to degree heating weeks is not good, the text doesn't relate what this means to the corals.

Line 217: This is not a surprise to the restoration community, which is how the paper reads from its framing. But if no action is taken at these sites, what would happen to the reefs. This balanced narrative should be considered.

Line 238: “Ties to accessibility” How was this determined or is it inferred?

Line 243: Is reference 65 for coral reefs? There are examples for reefs where you can have high diversity, eg lots of branching *Acropora* species but low function, and therefore resilience. Functional diversity of reefs should be discussed or the references changed.

Line 248: “Lack of consistency in how different projects assessed success”. Possibly, but not the only reason, e.g. maybe a predicted variable wasn't a driver of success. I suggest unpacking other options here.

Line 257-260: These recommendations seem to be a random selection from some comprehensive suggestions that have already been put forward in the community. And ultimately it depends on the purpose of the restoration program as to what monitoring can occur. Monitoring increases the costs associated with restoration, so for some community programs this won't be possible. Yes, it would be nice to have, but again trying to remove the local context is problematic.

Examples of guides:

Shaver, E., Courtney, C., West, J., Maynard, J., Hein, M., Wagner, C., Philibotte, J., MacGowan, P., McLeod, I., Bostrom-Einarsson, L. and Bucchianeri, K., 2020. A manager's guide to coral reef restoration planning and design.

<https://nesptropical.edu.au/wp-content/uploads/2021/02/NESP-TWQ-Project-4.3-Final-Report.pdf>

<https://icriforum.org/coralrestoration/>

Line 266: The database is a good idea, but what about issue with accessibility, particularly for local communities. These challenges and the potential for inequality to occur that can disadvantage communities that ultimately need reefs, should be recognised.

Line 289: And also functional diversity.

Line 289: What is meant by ecological reasoning?

Line 305-309: This section feels like a catch all caveat to try and balance the rest of the narrative. The title of the paper says restoration alone will not save coral reefs from climate change, but then here you say, "Our research does not answer this question" in relation to "whether coral restoration is an effective response to the ongoing coral decline". This statement is in contradiction to the whole narrative of the paper. I recommend adding balance throughout the narrative and then reworking this paragraph, so it is consistent with the messaging of the rest of the paper.

Line 340: Proximity to human settlement can hinder restoration success, ok, but there is no discussion again around the societal reasons why it could still be valuable from a socioecological perspective.

Line 349: What are the anthropogenic and climate stressors, please specific them.

Reviewer #3 (Remarks to the Author):

Review of Mula et al. – "Restoration alone will not save coral reefs from climate change".

This is an interesting paper. The authors carry out a series of analyses to highlight the current limitations and overall lack of feasibility of coral restoration efforts, but perhaps more interestingly attempt to identify what factors drive choice of restoration localities. Restoration is then placed in the context of future climate vulnerability – an important consideration for ecosystem restoration under climate change. These are all admirable aims, and I have no doubt this paper will spur important discussions.

Overall, the paper is generally well written, and I enjoyed reading it. However, there are several important things that need addressing to ensure the reader understands the approach in full. These include providing more details on the BRT modeling methods, showing us the underlying distributions in your data set (in particular the predictor variables), and in some cases discussing more about the potential limitations of your findings. I'm confident the authors can address my points of concern (even if some require additional analyses/information), which I outline in more detail below (in no particular order).

Major comments

1. BRT modeling.

There are several aspects of the BRT modeling methods missing which makes it hard to decipher exactly how your models were constructed and parameterized/optimized. Given these models provide some of your central results, the reader needs more detail. I know these methods well, and I am still slightly confused what your "variance" model performance measure actually represents.

First, you make no mention of the BRT "learning rate" or "tree complexity" parameter settings used anywhere in your methods. We need to know this. Importantly, we also need to know how you chose your parameter settings and whether these were consistent across your 100 BRT models? In previous research, we have varied the bag fraction (I'll come to this later below), learning rate and tree complexity to optimize our BRT models, re-running the routines with each unique combination of the parameter settings (normally something like: lr 0.01, 0.001, 0.0001; bag fraction 0.5, 0.7, 0.8; tree complexity 1-5) and identifying the parameter settings where predictive deviance is minimized. While computationally intensive, this has hugely improved the performance of our models in the past. Please either do this or explain why you did not need to optimize your models.

We also need to know the number of trees each BRT model consisted of, as a "BRT model" can each consist of 1000s of trees fitted in a stage-wise manner. This is important because in places (e.g. L391) you talk about aggregating the results of 100 BRTs, but the reader has no idea of the underlying complexity of each of these individual models. These kinds of summary stats, as well as summaries of your parameter setting mentioned above, need to be summarised in a Supplementary Table.

Please explain why you chose 100 iterations of your BRT models. Your data set is actually relatively small for BRT modeling, so I cannot imagine this was a result of computational limitations. It seems small and I would encourage you to see what happens if this is increased to 1000 or 10,000 iterations. Hopefully, if the signals are strong, the core results will remain, but some kind of sensitivity analysis like this would be reassuring to the reader.

You calculate the relative importance of your predictors in your BRT models, but you never actually tell us how you did this. I'm guessing it was something like calculating the number of times each was selected for splitting, weighted by the squared improvement to the model as a result of each split, and averaged over all trees (sensu Friedman & Meulman 2003)? Either way, please explain this process to the reader.

Friedman, J. H. & Meulman, J. J. Multiple additive regression trees with application in epidemiology. *Stat. Med.* 22, 1365-1381 (2003). <https://doi.org/10.1002/sim.1501>

Also, regarding the relative influence of the predictors, I would strongly encourage you to only interpret and plot those predictors with an importance greater than that expected due to chance alone (100/number of variables) – see Muller et al. (2013):

Müller, D., Leitão, P. J. & Sikor, T. Comparing the determinants of cropland abandonment in Albania and Romania using boosted regression trees. *Agricultural Systems* 117, 66-77 (2013).

What does your “deviance explained” (L364) represent? Is this more akin to the traditional R2 measure of model performance? If so, this is not where the power of BRT is. A more appropriate (and much more rigorous in my opinion) measure of model performance would actually be “predictive deviance explained”, i.e. the ability of your model to predict to data it has not seen before. You could do this using cross-validation given your relatively small data set size. By creating iterations of “test” and “training” data sets which would be set by your “bag fraction” (after your initial data re-sampling routine of 50% restored sites, 50% control sites) you could each time calculate the predictive deviance and then average this value across all iterations. This would give you a cross-validated predictive deviance explained as a measure of model performance, calculated as: $(1 - (\text{cross-validated deviance}/\text{mean total deviance}))$ (see Jouffray et al. 2019 – you can calculate this using the ggBRT routine written as part of this paper). My concern is that this CV-predictive deviance explained may be smaller than the “deviance explained” measure of model performance you report. Please calculate this more appropriate measure of model performance or explain why you think it is not necessary.

Jouffray, J. et al. Parsing human and biophysical drivers of coral reef regimes. *Proceedings of the Royal Society B: Biological Sciences* 286, 20182544 (2019). <https://doi.org/10.1098/rspb.2018.2544>

Finally, I would also ask that you superimpose your fitted values on the partial dependency plots, for full transparency. We’ve done this in the past using the ggPDdual_boot function in the ggBRT routine I mention above.

If I have misunderstood in any way and you have, in fact, done some of the things I request above, then this is likely an indication that your methods need to be more carefully articulated to the reader.

2. Poor performance of “short-term success” model.

Given the very poor performance of this model (13.2%) I think it is misleading to include Figure 3 in the main manuscript. The danger of it, as presented, is that the reader uses these kinds of plots in talks etc without the poor model performance context. I would strongly advise this figure is moved to Supp. Mat and the attention instead focused on the poor model performance and the inability to therefore read much into the resulting relationships whatsoever in the main manuscript.

I think you also need to mention a little more about the kinds of things that might have affected “short-term success” that you weren’t able to include as predictors in your models, mostly because those data layers don’t tend to exist for reefs and they are not often captured well by proxies like distance to people (e.g. degrees of wastewater pollution, urban runoff, sedimentation rates). See Gove et al. (2023) for an example of this in Hawaii – some of these coastal stressors are actually higher where there are fewer people along the coastline, because golf courses/plantations that cause coastal nutrient pollution/sedimentation inherently mean there are few people living along those portions of the coastline. As such, distance to major population center fails to capture the spatiotemporal patterns in these drivers of reef decline. There needs to be more discussion of the limitation of the proxies that you use as predictors in your model here I think.

Gove, J. M. et al. Coral reefs benefit from reduced land-sea impacts under ocean warming. *Nature* 621, 536- (2023)

3. Figure 4.

This is an interesting exercise, but we need to know these results in relation to their nearby non-restored (control) reefs, and also when it comes to looking to the future projections. Some locations/countries experience very low small-scale variability in SST relative to other locations. For example, Fiji and Northern French Polynesia have very low local-scale SST variability and the climate projections for these locations have similarly low local-scale variability (see van Hoodonk et al. 2016). As such, any restoration efforts in these locations will undoubtedly have a more spatially homogenous threat of bleaching (potentially very near in the future). This doesn’t mean, presumably, these locations should just be ignored for potential restoration and I think you need to discuss this a little more when it comes to the Discussion section of these results (and the future projection results) (L274-283).

Are you able to add the % control sites exposed to bleaching in each of the years on Fig. 4? I think this will provide important context. It would also be useful to see the number of thermal anomalies over time (for all reefs) on the plot as well. This will allow us to see whether the restored sites and control sites show similar patterns, and how these patterns relate to overall ocean warming trends.

4. Predictor variables in your models.

Given you have some predictors that can be measured very accurately at the scale of your response variable (e.g. distance to population center) and others that have much coarser spatial resolution and are likely to have a scale mis-match with your response variable (e.g. SST) this needs some discussion in the paper. How might this have affected your results and interpretation? The lack of strong relationships with some of your predictors could be due to this scale mis-match (rather than there being no inherent underlying relationship) and this warrants some discussion in the paper.

Please plot the distribution of ALL your predictor variables generated (I suggest as histograms) and included in your model-fitting process and provide as Supplemental Figures. At no point are the underlying predictor variable data shown to the reader (also see my later minor comment about adding the predictor variable distributions to the partial-dependency plots). While BRT can handle some wild data distributions, we still need to know if some of your predictors have very limited replication along their x-axes.

Why do you not investigate your predictor variables for collinearities before including them in the model fitting process? BRT is fairly robust to overfitting, however this is good practice in ecology as we are striving for model parsimony. Please calculate the pairwise correlations of all your predictor variables. If some are highly correlated (e.g. $r > 0.8$), I would advise removing one of each pair that is (prior to model fitting) and justifying this choice.

5. Climate exposure versus climate vulnerability.

In the Discussion section of your paper you talk about your results in the context of "climate vulnerability" (e.g. L274-275), however you do not actually measure this, you measure "climate exposure" and these are two very different things. SST thermal stress is a measure of "exposure" to future climate stress (i.e. the risk of being "exposed" to bleaching-inducing temperatures), but the "vulnerability" of these reefs is a combined measure of their climate exposure plus some measure of their resilience to stressors (both climate and local anthropogenic). You do not have the latter in your data. As such, please modify your language accordingly in this section.

I would also like to see a better discussion of how other local-scale factors (not measured here) could affect how the "restored reefs" do in the future (both anthropogenic and climatic/oceanographic). The title of your paper infers that you have local knowledge of reef resilience to climate change (beyond just exposure), but you do not. Your measure of climate exposure alone does not provide a conclusive proxy for how these reefs might fare under climate change. I would ask that you tone down the paper title to reflect this. I understand the paper title is eye-catching, but it's potentially false advertising and misleading.

6. Relatively low performance of your core model (restoration site location).

L153 – 42.1% is relative low model performance and means more than half the variation remained unexplained (and this value is potentially lower if you calculate the CV-predictive deviance explained as I request above). In this section, you need to dedicate more space to discussing what else might be driving the response variable that you missed or could not measure (due to those data layers not existing). There are likely complex geopolitical reasons for restoration site choice that go beyond "distance" and none of these social variables were included. This doesn't need to be lengthy, but you need to more directly acknowledge the large amount of variation that remained unexplained in your model and what might be required in the future to identify the influential social-ecological factors at play here. Numerous papers discuss the strong influence of social factors in dictating reef structure and function in recent decades (e.g. Norstrom et al. 2016, Hughes et al. 2017, Williams et al. 2019), and it is highly likely it is some of these social variables driving site choice.

Norström, A. V. et al. Guiding coral reef futures in the Anthropocene. *Front. Ecol. Environ.* 14, 490-498 (2016).

Hughes, T. P. et al. Coral reefs in the Anthropocene. *Nature* 546, 82 (2017).

Williams, G. J. et al. Coral reef ecology in the Anthropocene. *Funct. Ecol.* 6, 1014-1022 (2019).

Minor comments

L30 – "comprehensive dataset". This is vague – please give overall replication in abstract as this is central to the paper's rigour.

L36 – "near future" – please quantify this statement.

L39 – "special attention to future climate conditions" – bit vague. You focus on ocean warming trends, so why not just say that or at least provide more context here as it is the final sentence of your abstract (and arguably, therefore, one of the most important sentences in your entire paper).

L45 – I would encourage you to also look at a much more recent paper on coral reef ecosystem services by Woodhead et al. (2019): <https://besjournals.onlinelibrary.wiley.com/doi/full/10.1111/1365-2435.13331>

L45-46 – I think you need to more directly state "and local human stressors" here and give some key examples. Some good

empirical examples (not review/opinion papers) of local stressors comprising reefs in the context of climate change/bleaching are as follows:

Gove et al. (2023): <https://www.nature.com/articles/s41586-023-06394-w>

Graham et al. (2015): <https://www.nature.com/articles/nature14140>

L49-50 – Need more balanced argument here given the relatively small number of papers out there on this topic. I would draw your attention to Fox et al. (2021) who showed quite the opposite point to the one you are making:

<https://agupubs.onlinelibrary.wiley.com/doi/full/10.1029/2021GL094128>

L64 – “local scale” – please quantify here what you mean by this, it’s subjective.

L73-74 – Again need to mention here that coral loss following heat stress is often exacerbated by local human stressors.

Gove et al. (2023) and Graham et al. (2015) both relevant here again, but so is Donovan et al. (2021):

<https://www.science.org/doi/full/10.1126/science.abd9464>

L103-105 – Please elaborate on what you mean by “overall condition” and give some examples. I assume one is habitat complexity, which is often not re-gained even in restored systems (e.g. intertidal saltmarshes:

<https://www.sciencedirect.com/science/article/pii/S0925857418300260>).

L106-110 – You don’t mention recent paper by Lange et al. (2024) about measuring carbonate budgets of “restored” reefs that seems highly relevant here as this at least tries to look in more detail at the habitat attributes of “restored” reefs:

[https://www.cell.com/current-biology/fulltext/S0960-9822\(24\)00151-9](https://www.cell.com/current-biology/fulltext/S0960-9822(24)00151-9). This paper is also relevant in L116-120.

L114-115 – It would seem relevant here to also mention that some agencies don’t even actually monitor for success or not, see recent paper by Lamont et al. (2023): <https://www.science.org/doi/abs/10.1126/science.adh2610>

L123 “220 restoration” – change to “220 coral restoration”. Also, we need a map of these localities. I found it frustrating that I had to go to the paper you cite in your methods to find a map. Please either provide as a Supp. Fig, or consider adding it to Figure 2 as an additional panel above.

L148 – Why did you not use the “Gravity” metric previously developed for coral reefs by Cinner et al. (2018) which combines travel time with population size? This would seem a more refined method than an arbitrary cut-off population center size (see next comment). <https://www.pnas.org/doi/10.1073/pnas.1708001115>

L149 – “large” – please quantify. What was the threshold population density to be considered “large”?

L168 – Figure 2, we need to know the replication along the x-axis of each partial dependency plot. You could do this by adding deciles (sometimes called “rug plots”) to the top of each plot. Also need x-axis label on panel “a”.

L206 – “4 <” – shouldn’t this be the other way round “<4”)?

L208-210 – Please also report the % that experienced more than 8 DHW as this is a common threshold that causes severe and widespread bleaching (sensu Skirving et al. 2020):

Skirving, W. et al. CoralTemp and the Coral Reef Watch Coral Bleaching Heat Stress Product Suite version 3.1. Remote Sens. 12, 3856 (2020).

L215 – Is SSP2-4.5 the trajectory we are currently on? I didn’t think it was. Either way, please explain and justify to reader more why you chose this particular SSP projection.

L228 – please explain what “other reef localities” are in more detail in figure caption.

L232 – Figure 6 – the maps are too small. Suggest they are made full page width and put the box plots below.

L238 – change to “reef accessibility”.

L264 – a more appropriate term these days is “uncrewed” as it is gender neutral.

L271-273 – it would be useful context to refer some other database examples here that have been successful, for example Mermaid for reef monitoring: <https://datamermaid.org/>

L349 – Your choice of the term “environmental” here is perhaps misleading given you also include biological/ecological factors like coral diversity. I would suggest just using the term “factors” when referring to your predictor variables.

L516 – Latin name should be in italics.

*****END*****

Version 1:

Decision Letter:

6th January 2025

Dear Dr. Strona,

Thank you for submitting your revised manuscript "Restoration cannot be scaled up globally to save reefs from loss and degradation" (NATECOLEVOL-24061749A). It has now been seen again by the original reviewers and their comments are below. The reviewers find that the paper has improved in revision, and therefore we'll be happy in principle to publish it in Nature Ecology & Evolution, pending minor revisions to satisfy the reviewers' final requests and to comply with our editorial and formatting guidelines.

[redacted]

Reviewer #1 (Remarks to the Author): none

Reviewer #2 (Remarks to the Author):

Thank you to the Authors for taking the time to respond and edit their manuscript, in detail, to the points I raised in my first review. The paper reads much more balanced around the complexities that exist with reef restoration, and I happy to recommend publication in its current form. Well done again!

Reviewer #3 (Remarks to the Author):

I appreciate all the work the authors have put into addressing my concerns/suggestions in the initial review of this work (I was the original Reviewer 3). The statistical modelling approaches are now far clearer and robust, and it is now much easier for the reader to follow the logic of the approaches. There are, however, a few remaining things to iron out, so that graphics are streamlined and convey the critical piece of information, and that key methods steps are clearly defended and justified, as well as a series of other minor suggested edits for clarity.

1. I still think the title of the paper should focus more on the results at hand, rather than the overall 'Discussion' point of the piece. For example:

Coral reef restoration efforts are not guided by local human impacts or climate vulnerability

or

Coral reef restoration efforts are not guided by local and global human impacts

This is clearly something the editors can discuss with the authors.

2. Line 56 – this is slightly inaccurate. Gove et al. focus on 'reef-builder' recovery following bleaching-induced coral mortality (i.e. hard coral + crustose coralline algae). Suggest you re-phrase to "maintaining reef-builder cover (hard coral + CCA)".

3. Line 66 – "out-plantation" – should this be "outplanting"?

4. Line 82-84 - Are these really "regional-scales"? This implies intra-country scales. I thought these examples were much smaller? I would just state the actual restoration scale of these examples, rather than trying to generalise (which could be misleading). For example, the Lange et al paper is talking about 4-8 ha area only: "To date, the Mars Coral Reef Restoration Program (www.buildingcoral.com), has restored >4 ha of reef across two neighboring islands in the Spermonde Archipelago, South Sulawesi (and >8 ha across Indonesia in total)."

5. Line 134 "in specific occasions" is odd phrasing. Do you mean "under specific conditions"?

6. Line 137 – “such as reducing greenhouse gas emissions” – while I appreciate the importance of this, I do think the role played by local human land-sea impacts is also worthy of mention alongside this. Plenty of evidence of this driving reef decline (e.g. Gove et al. 2023).
7. Line 155 – “outcomes” – perhaps “interpretations” is more suitable here? I don’t see how you can have an inaccurate outcome....the outcome is the outcome.
8. Line 186 – remoteness from what? Need to define from what early on to the reader at this point.
9. Line 204 – define “large” here please for the reader, rather than them having to look in methods as this represents crucial context.
10. Line 211 – “devised” – perhaps “built” is more appropriate?
11. Line 221 – “more pristine”. There are no “pristine” reefs left, whatever that might mean, so I think you’re safer just saying “less impacted” here.
12. Figure 2 – while I appreciate your attempts to follow my request in the original review here, this is not what I was getting at, and I did in fact mean “ggPDdual_boot” function. This is confusing - what are the fitted trend lines in the ‘b’ panels then and why do they differ from in ‘a’ (noting you don’t explain what these are clearly in the figure legend)? My original suggestion was to add the fitted values to your fitted functions (i.e. add fitted values to your ‘a’ panels). As mentioned in my previous review, this can be done using the ggPDdual_boot function(I did not mean the ggPDfit function as you suggest in your response). We need the following three pieces of information all on each of the partial dependency plots: 1) fitted function(orange line in example figure below), 2) error around the fitted function(grey shading), 3) underlying fitted values (grey open circles) - see below example we built for an analysis. The ggPDdual_boot” function will help you do this, as I’m sure there are other functions perhaps by now too. Your PDfit plot is not useful, as it’s unclear what the fitted function in this represents (and why it is different from the fitted function from your model outputs in panels ‘a’). Attached is one we built for some drivers of water quality analyses. Note how each partial dependency plot has the 3 pieces of information I mention above on each plot (hopefully this helps clarify).
13. Line 234 – “technique/s” – should be “technique(s)”
14. Line 240 - It’s not clear to me how reference to Extended Data Figure 4 here helps this part of the story here. Please help the reader see the links more.
15. Figure 3 – I don’t see you refer to this figure anywhere in the main text. Please explain/fix. Need to help the reader understand why this plot is relevant to the story more in the main text.
16. Line 268-269 - But can’t an increase of ~1degC trigger bleaching depending on the number of DHW? I do not understand the motivation for this conservative approach. It would be useful to see both 1.0 (or 1.5) and 2.0 deg C (as a 2 x 2 panel plot), or more clearly defend/justify going against the standard NOAA protocol here.
17. Line 274-275 - Ok, but it’s more about the frequency here than the average timing of the anomaly isn’t it? Reefs can cope with being bleached (they have adapted to), but not if it happens too often. We want to know if anomalous temperatures will occur at such a frequency as to compromise recovery windows. This would be much more informative if expressed as ‘number of anomalies per decade’. It could then be plotted as a continuum rather than just quoting these mean timing values. See McWhorter et al. 2021 (Fig. 1b) for inspiration here: <https://onlinelibrary.wiley.com/doi/epdf/10.1111/gcb.15994>
18. Line 281 - Why do titles on graph indicate “I-II” and “II”...is this supposed to mean I to II and than II or greater? Not clear.
19. Line 304-306 - This question has been raised before and discussed in detail (please acknowledge this work: https://conbio.onlinelibrary.wiley.com/doi/abs/10.1111/j.1523-1739.2008.01037.x?casa_token=o-q1-q0aPZUAAAAA:JFIdz2_c1rA3IVHcwVlyAKmdqDQRA4cYzybo6OWp7ihyTVe6koZNw4UwTaA_-4aNosxW_j-u_Z5GcLc
20. Line 237-238 - This is a little too brief - please expand slightly for the reader. Observed inconsistency in what? You mean the poor performance of your model?
21. Line 377-379 - You cite 2 review papers here. Better to cite primary data papers for these kinds of statements, for example this paper that you already cite in Intro: <https://agupubs.onlinelibrary.wiley.com/doi/full/10.1029/2021GL094128>
22. Line 416 - The Gove et al. paper is very relevant again here as this shows you only achieve reef resilience to bleaching if both land and sea based stressors are mitigated simultaneously.
23. Line 486-488 - This is all much clearer now. However, these bag fractions of 0.8 are high. How does your model performance vary if you drop this down in increments of 0.1 to 0.5? I would prefer that you report the range in model performance values having done this sensitivity analysis. I know this is work, but I think that will be more reassuring to the reader and prevent folks worrying about overfitting here because of a high proportion of training data.
24. Line 488-489 - Please re-phrase for non-expert. What is the issue you are worried about and what is the goal you are

hoping to achieve with this approach? You simply report the 'method' without explaining the reasoning behind it.

25. Line 510 – “relative influence” - Please at least reference the used methods approach here (I appreciate you do not need to add detailed formula).

26. Line 510 – “marginal variable effects” - Aren't these conditional effects? i.e. the effect of a given predictor holding all other predictors at their mean values?

27. Lines 557-559 - Again - I think you need to make the motivation behind your approach/choice here very clear - why go against the standard NOAA approach? This will come as a surprise to readers and we need to know why you've deviated (you need to defend/justify this more I think).

Reviewer #1

I enjoyed the economic analysis of the costs of restoration at scale. This alone should cause anyone advocating restoration through direct intervention to carefully consider other options. I think it would be very helpful if the authors provided some additional information as to what is included by their referenced sources. For example, are the costs of assisted evolution (e.g., genetic engineering) and assisted gene flow included. Both are inordinately expensive, are nearly impossible to scale geographically, are applicable to a vanishingly small number of species, and nearly always do not properly consider the evolutionary dynamics that can impose considerable maladaptation in their application.

RESPONSE #1: Thank you for the positive feedback and valuable suggestions. We have clarified that the costs and scalability of advanced techniques such as assisted evolution and assisted gene flow remain uncertain, because they have not yet been tested at scale, and there are few studies on their broader application. We have added this clarification at L97–101. However, we have also now expanded the analysis of economic costs (see new Fig. 1) showing the hypothetical trajectories for different coral restoration techniques based on data from Bayraktarov et al. (2019).

I also would like to see the authors be a bit braver and conclude their paper with a set of next steps/best options for research and practice in this area for both coral reefs and beyond. For example, it is clear and I agree, they do not believe that direct intervention is likely to succeed without considerable modification to the methods being currently applied. Therefore, what about indirect intervention before CO₂ can be adequately drawn down such as say combinations of MPAs and water quality remediation that also benefits landholders. While beyond the scope of this paper to consider in depth, it might also be useful to include in this discussion, indirect interventions that will also fail such as the direct removal of Crown-of-Thorns starfish at scale.

RESPONSE #2: We have added a final remark to the conclusion to propose potential complementary strategies (L413–419):

“Considering the constraints facing current restoration methods, our findings underscore the importance of exploring indirect interventions to support coral resilience while addressing CO₂ reduction on a broader scale. For instance, combined approaches such as marine protected areas (Strain et al. 2019) and water-quality remediation (Nalley et al. 2021; Brodie et al, 2019) could create synergistic benefits for reefs, while also supporting local communities with incentives for conservation. Although beyond the scope of this paper, reinforcing complementary strategies could bolster ecosystem resilience, extending the reach of coral restoration.”

Reviewer #2

... I have serious concerns on how the paper has been framed, which fails to capture the reality of why restoration is occurring where it is, e.g., the societal element of coral reefs. Further, there are broad statements about restoration that cannot be generalised. For example, "Coral restoration is gaining popularity as a mitigation strategy". What it is apparently mitigating against was not specified, but time and time again in the coral literature, restoration is not described as a substitution for climate action. Restoration exists as a continuum of approaches from mitigation of a stress through to rehabilitation (see Suggett et al., 2024). This nuanced understanding of the intent of restoration is crucial, as the current framing of the paper reads as though it is trying to disprove restorations validity to mitigate climate change, which is not a universal narrative being put forward by the restoration community. Without a more nuanced discussion of restoration intent, I cannot recommend the work for publication.

RESPONSE #3: It was not our intent to convey a misleading message. We agree that restoration actions can have multiple goals at different temporal and spatial scales, and that understanding those goals is fundamental to ensure a proper evaluation of their effectiveness and success. We have now extensively reworked the manuscript to make it clear that **the main objective of our analyses is to explore the potential of coral restoration as a broad-scale (and long-lasting) solution.** Our message is now clearer and more nuanced.

Some of us authors are also directly involved in restoration projects. Our intent was by no means to criticize the coral-restoration community, especially because most scientists actively involved in the field have a clear vision of what restoration can and cannot do, as the reviewer stated. However, the narrative of restoration as a primary mitigation strategy is widespread in the media (e.g., theguardian.com/environment/2020/oct/17/why-there-is-hope-that-the-worlds-coral-reefs-can-be-saved; theguardian.com/environment/2021/may/16/could-engineered-coral-save-the-planets-reefs-from-destruction; nytimes.com/2017/09/20/climate/coral-great-barrier-reef.html), and so the prevailing message has important impacts at different societal and economic scales (e.g., new *Nature Restoration Law* in Europe: environment.ec.europa.eu/news/nature-restoration-law-enters-force-2024-08-15_en). As stated in Streit et al. (2024), it is alarming that the growing popularity of these interventions might (potentially) distract attention from taking direct and effective action to reduce greenhouse gases, undermine critical research on coral reef change, and place too much optimism about scientific and technological restoration capabilities. This is particularly concerning considering limited evidence that these interventions meaningfully improve the state of coral reefs, making it difficult to distinguish hopeful aspirations for change from robust ecological evidence of their

effectiveness. Our intent here is to stimulate debate on this important topic by tackling some overarching questions and hypotheses using a quantitative approach.

L134-139: “At the same time, the growing focus on restoration interventions risks diverting attention from addressing the root causes of coral reef decline, such as greenhouse gas emissions, and fostering optimism that may exceed the current evidence for their effectiveness (Streit et al. 2024)”

Line 25: “possible mitigation strategy” – mitigation to what? If climate action, please remove as this is not a common intent of restoration and fails to capture the continuum along which restoration exists (see Suggett et al. 2024).

RESPONSE #4: We now emphasize that coral restoration is part of a broader array of approaches aimed at rehabilitating compromised coral reefs, rather than solely a strategy for mitigating climate change (L25–30):

“Coral restoration is gaining popularity as part of a continuum of approaches aimed at addressing the widespread, recurring coral mass-mortality events that — together with elevated and chronic mortality, slower growth, and recruitment failure — threaten the persistence of coral reefs worldwide. However, the monetary costs associated with broad-scale coral restoration are massive, making widespread implementation challenging, especially with the lack of a coordinated and ecologically informed planning.”

Line 28: Clarify what the mass-mortality refers to.

RESPONSE #5: We have added “coral” to clarify the sentence.

Line 27: What is meant by cost? Societal, financial, ecological? Please clarify.

RESPONSE #6: Because we analyse the scalability of restoration costs, we mean *monetary* costs. We have now clarified this point (L30–32). Other ‘costs’ (e.g., societal, ecological) clearly exist, but they vary substantially depending on the specific settings and would be difficult to identify, let alone quantify:

“However, the monetary costs associated with broad-scale coral restoration are massive, making widespread implementation challenging, especially with the lack of a coordinated and ecologically informed planning.”

Line 27: “The costs of restoration are as massive as yearly costs” – This cannot be

generalised for all reefs and all restoration practices. These general statements should be removed or rephrased with specific, evidenced based examples.

RESPONSE #7: We have rephrased the statement to avoid generalizing, focusing instead on the monetary costs associated with the scalability of coral restoration, not coral restoration itself (see Response #6; L30–32):

Line 32: Is global planning even appropriate? Who would make these choices, and what about the social licence of local communities and their voice in decision making?

RESPONSE #8: We agree that the term ‘global planning’ might be misleading and possibly excessive. Our intended meaning is the need for a ‘coordinated’ effort that respects and incorporates the voices of local communities in the decision-making process. We have now reworked the sentence to clarify this, removing the word ‘global’ (see also Response #7).

Line 31: “We highlight how such a plan” – What is the plan? It is unclear.

RESPONSE #9: We now specify that we are referring to the “coordinated and ecologically informed approach” from the previous sentence (L32-35):

“By combining a comprehensive dataset documenting the success of coral restoration along with current and forecasted environmental, ecological and climate data, we highlight how such a coordinated and ecologically informed approach is far from being realized, despite the extent of previous and ongoing efforts.”

Line 34: Yes, but what about the flipside of the argument, that close to human settlement is where you have the highest social needs, and therefore desire and need to restore reefs. This is a crucial viewpoint that has been missed from the paper.

RESPONSE #10: This is a good point. We have now added a sentence in the Discussion about this aspect (L300–311):

“An additional explanation is that many initiatives arise locally, driven by immediate needs and socio-cultural priorities, leading to patterns that might lack a clear ecological rationale. Proximity to human settlements, while associated with greater degradation and human influence, offers socio-ecological benefits. This raises the question: is it better to focus restoration on a site where the need is moderate, but the likelihood of success is high, or on a site with high restoration need but low chances of success? While locally driven restoration can enhance socio-ecological resilience,

contribute to community livelihoods, and protect coastlines in the short term, these efforts may prove unsustainable and ineffective in the long run. Alternative interventions directly targeting these specific needs might be a safer and more effective strategy. Ultimately, although social and cultural dimensions are beyond the scope of our study, we recognize their importance for restoration success and emphasize the need for ecologically informed guidelines that can ultimately integrate local objectives.”

Our paper focuses on the ecological value and vulnerabilities of coral restoration sites. While the social needs and motivations near human settlements are indeed important, they fall outside the scope of our current analysis, which is centred on ecological impacts. We chose to prioritize this perspective to maintain a clear focus on the environmental aspects of restoration.

Line 38: “Influenced by relevant ecological and environmental predictors” – can an example be provided to provide clarity.

RESPONSE #11: We have now provided an example (i.e., cumulative impact).

Line 38: “to reinforce joint development of restoration guidelines that go beyond local objectives” Is this appropriate? What about the voice of local people in decision making and the local site-specific context that shapes why restoration is being undertaken, and therefore what success is.

RESPONSE #12: We have now added a clarification early in the paper that our study is focused on the ecological aspects of coral restoration and the importance of developing guidelines that consider future climate conditions. We have also emphasized that the voice of local communities and site-specific contexts are necessary to the broader discussion of restoration success, but that these social and cultural dimensions fall outside the scope of our current study. These changes clarify that our overarching goal is to emphasize the need for ecologically informed guidelines that can be integrated with local objectives in future work. See also Response #10 and L300–310.

Line 48: Update reference 2; it is 1 Billion people (Sing Wong ... (2022)

RESPONSE #13: We have now edited this value and updated the reference.

Line 53: Needs a reference.

RESPONSE #14: We have added the following reference: Hughes, T. P. et al. Global warming and recurrent mass bleaching of corals. *Nature* 543, 373-377 (2017).

Line 51-52: As per my comment in the abstract, this is not a fair representation of the continuum of approaches that restoration incorporates. Please balance this narrative to reflect this. Many restoration studies explicitly state that restoration is not a replacement for climate action, for example in Suggett et al. (2024), “Ultimately, there is no “one size fits all” method for restoring coral reefs or measuring success, nor will we restore our way out of the climate crisis”. Such balance should be provided in the paper.

RESPONSE #15: We have revised the text to clarify that coral restoration is part of a broader approach to addressing the climate crisis, rather than a standalone solution (L62–63):

“Coral restoration has been advocated as a promising tool to reduce coral loss and restore damaged reefs worldwide (Suggett et al. 2022; Williams et al 2019; Kleypas et al 2021; Duarte et al, 2020; Vardi et al 2021) as part of a broader approach to counteract the climate crisis (Suggett et al 2024).”

Line 60: There is a lot of literature since the 2020 Boström-Einarsson et al. work discussing the current state of reef restoration which is relevant here. This is summarised again in the Suggett et al. (2024), where they highlight that the life stage of restoration (eg science and R and D) relative to industrial application at scale are being incorrectly compared to then decide restoration has failed. To work out the best approach, fast-fail science is being applied, and this is a necessary path if any scalable solutions are to be identified. Acknowledgement of this point should be included.

RESPONSE #16: We have now made this concept clear by adding the following (L72–79):

“However, an estimated 30–40% of all coral reef-restoration projects fail due to poor planning, unrealistic objectives, inadequate regular maintenance, and persistent anthropogenic pressures (Boström-Einarsson et al. 2021). This failure rate also includes trial-and-error projects, which are intentionally designed to test and improve restoration practices. These experimental projects play an important role in advancing restoration ecology by allowing researchers to refine techniques, identify challenges early, and test new methods under controlled conditions. Although such trials might not yield immediate restoration success, they provide insights that can improve future projects and contribute to a more robust understanding of the risks and limitations inherent in different approaches (Suggett et al. 2024).”

We have also added the requested reference.

Line 64: Could also include examples of restoration where reef function has been returned, for example calcification: Nuñez Lendo, et al. (2024); Lange et al. (2024).

RESPONSE #17: We have added the suggested examples (L81–86):

“There are emerging examples where reef function has been successfully restored at regional scales. For instance, recent studies have shown that coral restoration can drive rapid recovery of reef carbonate budgets following disturbances (Lange et al. 2024; Nuñez Lendo et al. 2024). These cases provide valuable insights into the potential for restoration to contribute to the recovery of reef functions, although the broader applicability and future scalability of these outcomes still needs further exploration (Hughes et al. 2025).”

Line 73: Other examples of restoration with much lower costs not included, eg Scott et al. 2024

RESPONSE #18: We have added this as an example of cost-effective restoration (L103–105):

“However, some lower-cost examples exist, such as tourism-led coral planting on the Great Barrier Reef, where leveraging existing infrastructure reduced costs to as little as US\$2.34 coral⁻¹ per trip (Gove et al. 2023; Scott et al. 2024; Graham et al. 2015; Donovan et al. 2021).”

Line 73: It would be balanced to also consider the cost of no action to local communities that rely on healthy, functioning coral reefs. Figure 1. It would be balanced to show what the loss of value would be if a reef is lost. See the example considered of value returned relative to restoration undertaken in Suggett et al. 2023

RESPONSE #19: We have now cited the suggested paper, and we have emphasized the value of reefs globally (in terms of ecosystem services) and how restoration might potentially bring not only environmental/ecological benefits, but also economic ones.

However, there is another fundamental aspect to consider. While the costs of restoration are certain, the benefits (especially in terms of returned economic value) are conditional on (long-term) restoration success, as well as on the relative contribution of restoration actions to health/survival of local reefs. An important part of our study highlights the high risk that climate change will jeopardize most restoration efforts. Our graph assumes a 100% success of restoration attempts, with the intended purpose

of showing the magnitude of the challenge of scaling up restoration under the most optimistic assumptions. There might be cases where local revenues (e.g., tourism) stemming from coral restoration actions might be substantial, but quantifying this kind of information globally is clearly unfeasible.

In principle, our plot does provide a frame of reference which might be used to quantify how much economic benefits one should derive from restored reefs to balance the costs (and possibly make restoration profitable). According to this reasoning, a restored reef should provide between 6,000 and 143 million US\$/ha⁻¹ (within a reasonable timeframe, e.g., a few years) to balance costs. But this kind of reasoning might be dangerous, because it could underestimate environmental and biodiversity aspects that are difficult or impossible to quantify (L89-92):

“Although reefs provide ecosystem services of immense ecological and economic value, a critical aspect to consider is that restoration costs are variable but quantifiable, whereas the benefits are complex, and conditional on long-term success and the specific contribution of restoration to local reef health and resilience.”

and L117–119:

“These figures highlight the vast monetary commitment required and the necessity for economic returns—ranging between US\$6,000 and US\$261 million per hectare—within a reasonable timeframe.”

Line 95: Provide examples of where restoration has been successful locally.

RESPONSE #20: We have now provided a few examples of successful restoration (see Response #17; L81–86):

Line 96: Same comment about mitigation being used here inappropriately.

RESPONSE #21: We have now removed the word ‘mitigation’ and edited the sentence (L133–135):

“This does not imply that coral restoration is ineffective at the local scale and in specific occasions, but it casts doubts on its scalability and its role within the broader toolbox of coral restoration strategies.”

Lines 100-106: These challenges have been highlighted in prior studies that should be acknowledge, e.g. the challenges being put forward are not newly described challenges.

RESPONSE #22: We have revised the text to acknowledge that these challenges are not new and have been highlighted in previous studies to provide proper context (L132–133):

“These challenges have been recognized in previous studies (Suggest et al. 2024; Boström-Einarsson et al 2020), which highlight similar concerns regarding the scalability of coral restoration.”

Line 125: What is the definition of success here? And how was this standardised for the 220 studies investigated. If one program had success as upskilling local communities and another was to increase biodiversity, how were these differences considered?

RESPONSE #23: We now specify in this paragraph that success is defined as growth and survival rate (‘ecological success’, L162). For our analysis, we focused on survival to standardize the evaluation across the 220 restoration projects (see L177–178).

Line 133: Is the data available?

RESPONSE #24: Yes. We have doublechecked and improved the data availability statement (L562–564). Specifically, the data for the global distribution of coral reefs are available from UNEP-WCMC. The coral restoration data are publicly available from Boström-Einarsson et al. (2020). Data on thermal anomalies are available from NOAA Coral Reef Watch. We retrieved information on coral diversity for each location from the coral species range provided by the IUCN (iucnredlist.org). Data on stressors are available from Halpern et al. (2019).

Line 134: “We hypothesise that some of these variables” – Please specify which ones.

RESPONSE #25: Done (see L186–187: “... i.e., remoteness, coral diversity, cumulative impact, bleaching events”)

Line 141-145: But what about the societal aspect of reefs. This is a missing component of the narrative you have for restoration site selection, yet it is key as reefs are socioecological systems.

RESPONSE #26: As we described in Responses #3 and #6, we agree that societal aspects could be important in determining the need and implementation of restoration actions. We now clarify that the spatial heterogeneity of local needs/societies/cultures

might explain a consistent portion of the variance left unexplained by our model (that aims to explore the role and importance of general ecological and environmental factors driving the selection of restoration sites at the global scale). See L300–311: “An additional explanation ...”

Line 199: What are the 14 stressors?

RESPONSE #27: We now provide a list of the stressors, L200-203 (i.e., commercial demersal destructive fishing, commercial demersal non-destructive high and low by-catch fishing, pelagic high and low by-catch fishing, artisanal fishing, sea surface temperature increase, ocean acidification, sea level rise, shipping, nutrient pollution, chemical pollution, direct human impact, and light).

Line 177: Without defining success and standardising this across the 220 studies, how can the drivers of restoration success be determined?

RESPONSE #28: We had explained this aspect in the Methods, but we were probably not as clear as we could have been. We have now explained how we standardized success as survival at the end of the monitoring time based on the information available in the dataset (L514–515):

“We combined information on the duration of the post-restoration monitoring and on the relative survival of the monitored colonies to obtain a standardized proxy for restoration success.”

We have also added a brief explanation in the main text (L177–178):

“For the scope of this study and based on the information available in the dataset used, we define success as survival relative to the monitoring time.”

Line 178: What are the 9 indicators?

RESPONSE #29: We now clarify that the indicators (eight in the new analysis we conducted) we used in the BRT model include the restoration technique, remoteness, gravity, local coral diversity, average and trend of cumulative impacts, pre- and post- (± 5 years) exposure to thermal anomalies (L233-237).

Line 187: Please clarify with higher or lower coral diversity.

RESPONSE #31: After doing the new analysis suggested by Reviewer #3, we changed the paragraph “Correlates of short-term coral restoration success ...” (L231-243).

Line 200-209: Is there information on mortality. What about narrative on environmental memory and the fact that some heat exposure can increase tolerance to future stress. While I agree exposure to degree heating weeks is not good, the text doesn't relate what this means to the corals.

RESPONSE #32: We have now incorporated additional information in the Discussion to clarify this (L377–384):

“There is growing evidence that repeated exposure to sublethal heat stress can increase coral resilience, a phenomenon referred to as ‘environmental memory’ (Hackerott et al. 2021; Brown & Barott 2022). This suggests that under certain conditions, corals might develop greater tolerance to future thermal stress. However, while our understanding of this mechanism and its trade-offs with other ecological processes is still limited, exposure to degree heating weeks above the critical thresholds (e.g., bleaching alert level II) continues to be a major concern, often followed by high coral mortality, especially in the absence of sufficient recovery periods between stress events (Hughes et al. 2017).

Line 217: This is not a surprise to the restoration community, which is how the paper reads from its framing. But if no action is taken at these sites, what would happen to the reefs. This balanced narrative should be considered.

RESPONSE #33: We have now added a section to the Discussion highlighting the importance of taking a more careful and strategic approach to restoration (L372–373):

“While this might not surprise some in the restoration community, it underscores the need for a more careful and strategic approach to restoration.”

Line 238: “Ties to accessibility” How was this determined or is it inferred?

RESPONSE #34: We now clarify that we inferred this from the boosted regression tree model that we devised to search for potential determinants for the selection of restoration actions (L294–295):

“Our boosted regression tree models revealed clear predictors for restoration site selection which also correlated with success, with reef accessibility being a key factor..”

Line 243: Is reference 65 for coral reefs? There are examples for reefs where you can have high diversity, eg lots of branching Acropora species but low function, and therefore resilience. Functional diversity of reefs should be discussed or the references changed.

RESPONSE #35: We have now added a reference specific to coral reefs (Bellwood et al. 2004), L300.

Line 248: “Lack of consistency in how different projects assessed success”. Possibly, but not the only reason, e.g. maybe a predicted variable wasn't a driver of success. I suggest unpacking other options here.

RESPONSE #36: We have now expanded the section discussing additional options besides lack of consistency in how projects measured success, such as the possibility that some of the predicted variables might not have been the most influential drivers of success, as well as the effects of unmeasured factors such as site-specific management practices, local disturbances, and ecological interactions (L317–328):

“However, we cannot exclude the notion that other unmeasured variables (e.g., site-specific management practices, local disturbances, or unforeseen ecological interactions) also determined restoration success. Furthermore, the varying spatial resolutions of our predictors likely influenced our ability to detect relationships. For example, predictors like sea surface temperature often have a coarser spatial scale that might not align precisely with the localized conditions of restoration sites, in contrast to more localized measurements such as distance to population centres. This scale mismatch could contribute to the lack of strong relationships with certain predictors not showing the expected associations because of spatial inconsistencies that obscure them. The large variation in restoration techniques and environmental settings adds further complexity, making it difficult to draw general conclusions about the overarching drivers of success. Additionally, the differences in coral species targeted and the varying local anthropogenic impacts might have contributed to the observed inconsistency.”

Line 257-260: These recommendations seem to be a random selection from some comprehensive suggestions that have already been put forward in the community. And ultimately it depends on the purpose of the restoration program as to what monitoring can occur. Monitoring increases the costs associated with restoration, so for some community

programs this won't be possible. Yes, it would be nice to have, but again trying to remove the local context is problematic. Examples of guides: Shaver ...

RESPONSE #37: We have tried to organize the section better and clarify its goals. We concede that some of our recommendations were not new, and it was our intention to focus instead on a few fundamental and specific needs emerging from our quantitative analyses rather than compile an extensive (and more generic) list. We have now emphasized this aspect (L338–351):

“To document restoration actions in line with our findings, we highlight a few fundamental metrics emerging from our quantitative analyses. These include a detailed characterization of the restored area, ideally incorporating relevant environmental and ecological parameters to establish a reference baseline for evaluating restoration outcomes. For coral-gardening initiatives, recording data such as the number of transplanted colonies per species, alongside their survival and growth rates at the target and adjacent control sites, would offer valuable insights. Regular assessments over a standard period would enhance the reliability of these metrics. Quantifying human intervention through comparable measures such as person-hours or estimated costs for activities like structure maintenance and algal removal, for example, would also aid in assessing resource requirements. We acknowledge that the extent of monitoring is contingent on the specific objectives of each restoration program and local resource availability. We focus here on highlighting essential metrics that would improve comparability and effectiveness of restoration at a broader scale. In line with recent guidance documents (Shaver et al. 2020), our suggestions are not exhaustive, but they do reveal the data needed to streamline monitoring and improve standardization.”

Line 266: The database is a good idea, but what about issue with accessibility, particularly for local communities. These challenges and the potential for inequality to occur that can disadvantage communities that ultimately need reefs, should be recognised.

RESPONSE #38: We have added a paragraph to acknowledge the challenges related to accessibility, particularly for local communities. In this new section we discuss the potential for inequality and emphasize the importance of designing the database to be inclusive and accessible to all stakeholders. We also suggest possible solutions, such as multilingual interfaces, offline data-collection options, and capacity-building efforts, to ensure that local communities can contribute to and benefit from this resource (L362–366):

“However, accessibility is essential for ensuring widespread use and engagement with such a database. To maximize inclusivity, the resource should be openly accessible, user-friendly, and equipped with multilingual

interfaces. By doing so, we can support diverse stakeholders involved in restoration planning and implementation, empowering them to contribute to and benefit from coral restoration efforts around the globe.”

Line 289: And also functional diversity.

RESPONSE #39: Added (L390).

Line 289: What is meant by ecological reasoning?

RESPONSE #40: We have now clarified that our term ‘ecological reasoning’ refers to the consideration of factors such as the target species’ role in ecosystem processes, their resilience to environmental stressors, and their capacity to support biodiversity and promote long-term reef recovery (L389–393):

“However, in most cases, the choice of the target corals is limited in terms of taxonomic and functional diversity, and not supported by ecological reasoning. This means that the selection of coral species often does not consider factors such as their role in ecosystem processes, their resilience to environmental stressors, or their capacity to support biodiversity and promote long-term reef recovery.”

Line 305-309: This section feels like a catch all caveat to try and balance the rest of the narrative. The title of the paper says restoration alone will not save coral reefs from climate change, but then here you say, “Our research does not answer this question” in relation to “whether coral restoration is an effective response to the ongoing coral decline”. This statement is in contradiction to the whole narrative of the paper. I recommend adding balance throughout the narrative and then reworking this paragraph, so it is consistent with the messaging of the rest of the paper.

RESPONSE #41: The modifications we described so far (e.g., Responses #17 and #18) provide a more balanced narrative. We have reworked this section, and we concede that it could have been misinterpreted. We now clarify that although coral restoration can be valuable in certain contexts, our findings suggest that it is not currently feasible to scale it sufficiently to have a meaningful, long-term impact on coral reef ecosystems. The revised paragraph also clarifies that restoration should be seen as one part of a broader strategy to address climate change and other synergistic threats to coral reefs (L406–412):

“All the challenges associated with planning and evaluation and those related to costs and scalability (Fig. 1) raise important questions about the current

effectiveness of coral restoration as a viable response to the ongoing coral decline (Bellwood et al. 2019). Our analyses suggest that while coral restoration has the potential to be a valuable tool in certain circumstances, it is not yet feasible to scale it up sufficiently to have meaningful, long-term, and positive effects on coral reef ecosystems. This reality check should stimulate constructive debate about maximizing the utility of restoration efforts, particularly in combination with broader strategies, to address climate change and other threats.”

Line 340: Proximity to human settlement can hinder restoration success, ok, but there is no discussion again around the societal reasons why it could still be valuable from a socioecological perspective.

RESPONSE #42: We have now elaborated on the socio-ecological value of restoration near human settlements in the Discussion (L300–310; see Response #10) and specify in the methods that (L453–454):

“Proximity to large human settlements can have the opposite effects of hindering restoration success through increased disturbances and simplifying operations for implementation and maintenance of restoration activities”.

However, restoration success is an even more important variable to consider than restoration need when planning. This boils down to the question: is it better to attempt restoring reefs in a locality where the need is intermediate, but the probability of success is high, or in a locality with high need for restoration, but where the chances of success are low? In the latter case, depending on the ecosystem services the reefs provide (e.g., fish protein, income from tourism, coastal protection, ...), alternative forms of intervention directly tackling the needs might represent a safer and more rewarding strategy.

Line 349: What are the anthropogenic and climate stressors, please specific them.

RESPONSE #43: See Response #27; we now list the stressors (i.e., commercial demersal destructive fishing, commercial demersal non-destructive high and low by-catch fishing, pelagic high and low by-catch fishing, artisanal fishing, sea surface temperature increase, ocean acidification, sea level rise, shipping, nutrient pollution, chemical pollution, direct human impact, and light; L469–472).

Reviewer #3

There are several aspects of the BRT modeling methods missing which makes it hard to decipher exactly how your models were constructed and parameterized/optimized. Given these models provide some of your central results, the reader needs more detail. I know these methods well, and I am still slightly confused what your “variance” model performance measure actually represents.

First, you make no mention of the BRT “learning rate” or “tree complexity” parameter settings used anywhere in your methods. We need to know this. Importantly, we also need to know how you chose your parameter settings and whether these were consistent across your 100 BRT models? In previous research, we have varied the bag fraction (I’ll come to this later below), learning rate and tree complexity to optimize our BRT models, re-running the routines with each unique combination of the parameter settings (normally something like: lr 0.01, 0.001, 0.0001; bag fraction 0.5, 0.7, 0.8; tree complexity 1-5) and identifying the parameter settings where predictive deviance is minimized. While computationally intensive, this has hugely improved the performance of our models in the past. Please either do this or explain why you did not need to optimize your models.

RESPONSE #44: We have now rerun all models by first searching the parameter space search as suggested, varying learning rate, bag fraction, and tree complexity (using the combinations suggested, with 10 replicates per parameter combination); and then generating 1000 models (for the evaluation of model accuracy) using the setup leading to the highest average accuracy. Within each parameter combination we generated multiple models by progressively increasing the number of trees per model from 50 to 10000, with incremental steps of size 10, and then selecting the model minimizing the holdout deviance (see Response #45 and Methods). As we explain in more detail below (Responses #46 and #49), we have also improved the evaluation of model accuracy by quantifying it through cross-validation (also controlling for spatial autocorrelation; see Methods). After cross validation, we generated a final model from the full dataset, which was used to explore variable importance and marginal effects.

We also need to know the number of trees each BRT model consisted of, as a “BRT model” can each consist of 1000s of trees fitted in a stage-wise manner. This is important because in places (e.g. L391) you talk about aggregating the results of 100 BRTs, but the reader has no idea of the underlying complexity of each of these individual models. These kinds of summary stats, as well as summaries of your parameter setting mentioned above, need to be summarised in a Supplementary Table.

RESPONSE #45: We now specify that we used an R function (*gbm.step* from the `dismo` library) that builds (and cross-validates) models with an increasing number of trees (50, 60, ..., 10000), and then generates a model using the number of trees maximizing the holdout deviance (

14/topics/gbm.step). We used that approach at all stages of the workflow (i.e., in the parameter search, cross-validation and in the final model).

Please explain why you chose 100 iterations of your BRT models. Your data set is actually relatively small for BRT modeling, so I cannot imagine this was a result of computational limitations. It seems small and I would encourage you to see what happens if this is increased to 1000 or 10,000 iterations. Hopefully, if the signals are strong, the core results will remain, but some kind of sensitivity analysis like this would be reassuring to the reader.

RESPONSE #46: The results were indeed stable across replicates, which is why we deemed 100 models sufficient for the analysis. However, we have now included 1000 models in the cross validation, and then relied on a single complete model for the exploration of variable importance. In all models and at all stages of the modelling workflow (parameter selection, cross validation, final model),

You calculate the relative importance of your predictors in your BRT models, but you never actually tell us how you did this. I'm guessing it was something like calculating the number of times each was selected for splitting, weighted by the squared improvement to the model as a result of each split, and averaged over all trees (sensu Friedman & Meulman 2003)? Either way, please explain this process to the reader.

RESPONSE #47: We now specify that we computed variable importance as ‘relative influence’ described in Friedman (2001). In brief, relative influence I is defined as the relative influences of the individual inputs x_j on the variation of the function $\hat{F}(x)$, the latter being the function that maps the explanatory variables x to the response variable y . For a collection of M decision (regression) trees $\{T_m\}_1^M$, the squared influence \hat{I}^2 is calculated as:

$$\hat{I}^2 = \frac{1}{M} \sum_{m=1}^M \hat{I}_j^2(T_m),$$

where $\hat{I}_j^2(T) = \sum_{t=1}^{J-1} \hat{i}_t^2(v_t = j)$, which is the summation over the nonterminal nodes t of the J -terminal node tree T , v_t is the splitting variable associated with node t , and \hat{i}_t^2 is the corresponding empirical improvement in squared error resulting from the split. These squared influences then sum to 100 over all x explanatory parameters, which is analogous to the percentage of variation in the response variable y explained by each parameter.

We do not think this kind of detail is required in the manuscript, but we will follow the editor’s direction here.

In addition to that method, we tried another approach where independent variables are randomized one at a time and the associated reduction in predictive performance is computed (as in Breiman’s random forest variable importance; Breiman 2001),

computed using the entire training dataset. The two methods yielded similar results, so we eventually settled on Friedman's (2001) approach (because it is a better and more well-established procedure).

Also, regarding the relative influence of the predictors, I would strongly encourage you to only interpret and plot those predictors with an importance greater than that expected due to chance alone (100/number of variables) – see Muller et al. (2013).

RESPONSE #48: We are not convinced this suggested approach is valid. The suggested threshold of 100/number of variables in principle identifies the expectation of variable importance assuming that all variables are equally important. We can imagine a model based only on two variables (e.g., precipitation and temperature), both providing a substantial contribution to predictions. We can imagine, for instance, that including both variables gives a cross-validated true skill statistic = 0.7, while including only precipitation gives 0.3, and including only temperature gives 0.4. In such a situation, the relative influence of precipitation will be slightly smaller than that of temperature (let us say 40 for precipitation and 60 for temperature). According to the suggested approach, one should interpret only partial dependency of temperature, and consider precipitation “not significant” (less important than predicted by chance), which is clearly not the case.

We have read the description of the procedure in Müller et al. (2013), and could not find any reference supporting the approach, nor is it in the original paper/s on partial dependency (Friedman 2001). We noticed that the criterion is also used in the R package `ggBRT` and attributed to Müller et al. 2013, which seems therefore to be the only/original source for the approach. That is a bit risky, because it might popularize an approach that is not robust or justified from conceptual and statistical perspectives.

The partial-dependency plots do include relative variable importance, provided by the range of variation in the response variable (e.g., Greenwell et al. 2018). Reporting a partial dependency plot for a variable with low importance is not deceiving (especially if the scale of the y axes is kept constant across variables), because it simply shows that the target variable has little effect on the variation of the prediction. Therefore, we will keep all variables in the partial dependency plots. To emphasize and distinguish their relative importance, we have now fixed the y scale and reported the relative importance score within each plot.

What does your “deviance explained” (L364) represent? Is this more akin to the traditional R2 measure of model performance? If so, this is not where the power of BRT is. A more appropriate (and much more rigorous in my opinion) measure of model performance would actually be “predictive deviance explained”, i.e. the ability of your model to predict to data it has not seen before. You could do this using cross-validation given your relatively small data

set size. By creating iterations of “test” and “training” data sets which would be set by your “bag fraction” (after your initial data re-sampling routine of 50% restored sites, 50% control sites) you could each time calculate the predictive deviance and then average this value across all iterations. This would give you a cross-validated predictive deviance explained as a measure of model performance, calculated as: $(1 - (\text{cross-validated deviance}/\text{mean total deviance}))$ (see Jouffray et al. 2019 – you can calculate this using the ggBRT routine written as part of this paper). My concern is that this CV-predictive deviance explained may be smaller than the “deviance explained” measure of model performance you report. Please calculate this more appropriate measure of model performance or explain why you think it is not necessary.

RESPONSE #49: We have now updated the analyses following the recommendation, implementing cross validation in both models (i.e., the one for restoration site selection and the one for restoration success). We have also differentiated the accuracy evaluation between the two models. Specifically for the restoration site-choice model (where the response variable is binary), we are now quantifying accuracy based on true skills statistics and type I/II error rates. For the model predicting restoration success (a continuous variable), we are using goodness of fit (R^2) of predicted *versus* observed success in the testing sets. Despite the substantial changes in the approach, the results are similar to those in the first version. Also, as anticipated, the model calibration procedure improved the accuracy in the restoration site-choice model.

Finally, I would also ask that you superimpose your fitted values on the partial dependency plots, for full transparency. We’ve done this in the past using the ggPDdual_boot function in the ggBRT routine I mention above.

RESPONSE #50: Done (although we think you meant the function ggPDfit).

If I have misunderstood in any way and you have, in fact, done some of the things I request above, then this is likely an indication that your methods need to be more carefully articulated to the reader.

RESPONSE #51: Model validation is now much more robust in our revised version.

Given the very poor performance of this model (13.2%) I think it is misleading to include Figure 3 in the main manuscript. The danger of it, as presented, is that the reader uses these kinds of plots in talks etc without the poor model performance context. I would strongly advise this figure is moved to Supp. Mat and the attention instead focused on the poor model performance and the inability to therefore read much into the resulting relationships whatsoever in the main manuscript.

RESPONSE #52: Yes. We agree that variable importance is not necessarily reliable for a model explaining a small proportion of the total variance (although to be fair, anything > 10% variance explained in ecology can arguably be considered ‘reasonable’ given the chaotic nature of natural systems, but that is an argument for another time). We have therefore now replaced Fig. 3 with a scatterplot showing the low predictive ability of the model, and we no longer compute, report, or discuss variable importance anymore for this model.

I think you also need to mention a little more about the kinds of things that might have affected “short-term success” that you weren’t able to include as predictors in your models, mostly because those data layers don’t tend to exist for reefs and they are not often captured well by proxies like distance to people (e.g. degrees of wastewater pollution, urban runoff, sedimentation rates). See Gove et al. (2023) for an example of this in Hawaii – some of these coastal stressors are actually higher where there are fewer people along the coastline, because golf courses/plantations that cause coastal nutrient pollution/sedimentation inherently mean there are few people living along those portions of the coastline. As such, distance to major population center fails to capture the spatiotemporal patterns in these drivers of reef decline. There needs to be more discussion of the limitation of the proxies that you use as predictors in your model here I think. Gove, J. M. et al. Coral reefs benefit from reduced land-sea impacts under ocean warming. Nature 621, 536- (2023)

RESPONSE #53: We have now expanded the discussion on the potential reasons behind the low model performance (see Response #36; L316–318).

3. Figure 4. This is an interesting exercise, but we need to know these results in relation to their nearby non-restored (control) reefs, and also when it comes to looking to the future projections. Some locations/countries experience very low small-scale variability in SST relative to other locations. For example, Fiji and Northern French Polynesia have very low local-scale SST variability and the climate projections for these locations have similarly low local-scale variability (see van Hooijdonk et al. 2016). As such, any restoration efforts in these locations will undoubtedly have a more spatially homogenous threat of bleaching (potentially very near in the future). This doesn’t mean, presumably, these locations should just be ignored for potential restoration and I think you need to discuss this a little more when it comes to the Discussion section of these results (and the future projection results) (L274-283).

Are you able to add the % control sites exposed to bleaching in each of the years on Fig. 4? I think this will provide important context. It would also be useful to see the number of thermal anomalies over time (for all reefs) on the plot as well. This will allow us to see whether the restored sites and control sites show similar patterns, and how these patterns relate to overall ocean warming trends.

RESPONSE #54: Following this advice, we have now rethought the figure, which now reports the fraction of recently restored sites exposed to bleaching alert level I and/or II per year, compared to control (non-restored) sites. For each target year, we identified all the localities restored in the preceding five years and identified as a control all the remaining reef localities. Then we computed the fraction of restored and control localities that were exposed to bleaching alert levels I and/or II. The plots show data aggregated in 5-year intervals, with the numbers in the x axis indicating the upper boundary of the interval (i.e., 1990 = 1986–1990). We shifted the time series for the restored and control data horizontally by ± 1 unit to ease visualization. Dots indicate mean values, while arrows indicate standard deviation.

Given you have some predictors that can be measured very accurately at the scale of your response variable (e.g. distance to population center) and others that have much coarser spatial resolution and are likely to have a scale mis-match with your response variable (e.g. SST) this needs some discussion in the paper. How might this have affected your results and interpretation? The lack of strong relationships with some of your predictors could be due to this scale mis-match (rather than there being no inherent underlying relationship) and this warrants some discussion in the paper.

RESPONSE #55: We now discuss this aspect (L319–325):

“Furthermore, the varying spatial resolutions of our predictors likely influenced our ability to detect relationships. For example, predictors like sea surface temperature often have a coarser spatial scale that might not align precisely with the localized conditions of restoration sites, in contrast to more localized measurements such as distance to population centres. This scale mismatch could contribute to the lack of strong relationships with certain

predictors not showing the expected associations because of spatial inconsistencies that obscure them.”

Please plot the distribution of ALL your predictor variables generated (I suggest as histograms) and included in your model-fitting process and provide as Supplemental Figures. At no point are the underlying predictor variable data shown to the reader (also see my later minor comment about adding the predictor variable distributions to the partial-dependency plots). While BRT can handle some wild data distributions, we still need to know if some of your predictors have very limited replication along their x-axes.

RESPONSE #56: We now provide all the distributions in Extended Data Fig. 2 and Fig. 2.

Why do you not investigate your predictor variables for collinearities before including them in the model fitting process? BRT is fairly robust to overfitting, however this is good practice in ecology as we are striving for model parsimony. Please calculate the pairwise correlations of all your predictor variables. If some are highly correlated (e.g. $r > 0.8$), I would advise removing one of each pair that is (prior to model fitting) and justifying this choice.

RESPONSE #57: Yes, we did. We found weak pairwise correlations among all variables. The only partial exception was the post restoration number of bleaching alert events, where the correlation with the post restoration mean bleaching alert level was 0.64. We tried models both including and excluding the variable, finding that including it slightly increased accuracy (because, as the reviewer indicates, tree-based models such as boosted regression trees or random forest are not overly sensitive to collinearity among variables. Thus, we kept all the variables. However, we now provide correlation plots as Extended Data Figures 5 and 6.

In the Discussion section of your paper you talk about your results in the context of “climate vulnerability” (e.g. L274-275), however you do not actually measure this, you measure “climate exposure” and these are two very different things. SST thermal stress is a measure of “exposure” to future climate stress (i.e. the risk of being “exposed” to bleaching-inducing temperatures), but the “vulnerability” of these reefs is a combined measure of their climate exposure plus some measure of their resilience to stressors (both climate and local anthropogenic). You do not have the latter in your data. As such, please modify your language accordingly in this section.

RESPONSE #58: We have modified the text accordingly, replacing ‘vulnerability’ with ‘exposure’(L367-384).

I would also like to see a better discussion of how other local-scale factors (not measured here) could affect how the "restored reefs" do in the future (both anthropogenic and climatic/oceanographic). The title of your paper infers that you have local knowledge of reef resilience to climate change (beyond just exposure), but you do not. Your measure of climate exposure alone does not provide a conclusive proxy for how these reefs might fare under climate change. I would ask that you tone down the paper title to reflect this. I understand the paper title is eye-catching, but it's potentially false advertising and misleading.

RESPONSE #59: The title of our manuscript reflects a change we made after the initial editorial assessment of the paper — the editor considered the original title we proposed (*International coordination needed to make any coral restoration meaningful*) too generic, but also thought that our revised title might too be able to be improved.

Therefore, instead of *Restoration alone will not save coral reefs from climate change*, we can propose the alternative *Restoration cannot be scaled up globally to save reefs from loss and degradation*. Ideally, we also want the concepts of standardization and local benefit to be reflected by the title, but we are limited by the character length permitted for *Nature Ecology and Evolution*. We have therefore focussed on the ‘scaling’ issues in the newly proposed title to reflect our main aim.

Although we agree that vulnerability and exposure are different concepts, in the case of reefs and bleaching there is overarching evidence of the relationship between exposure to thermal anomalies and mass coral mortality. This does not mean that for coral, exposure equals vulnerability. Still, it does suggest that the relationship between exposure and vulnerability is not linear (and most likely threshold-based — i.e., given enough exposure, mass mortality is expected for different systems regardless of differences in their vulnerabilities). In our future projections we are using a conservative criterion to quantify heat stress (twice the limit used by NOAA), identifying conditions where coral survival would be highly unlikely regardless of local differences in vulnerability.

Furthermore, under the assumption of a roughly Normal distribution of vulnerability among reefs, globally we might expect that over- and underestimation of mortality would compensate, which ensures that our global projections are reasonable (despite its quantitative nature, our paper does not aim to make predictions, but instead more scenario-based estimates to answer broader questions).

L153 – 42.1% is relative low model performance and means more than half the variation remained unexplained (and this value is potentially lower if you calculate the CV-predictive deviance explained as I request above). In this section, you need to dedicate more space to discussing what else might be driving the response variable that you missed or could not

measure (due to those data layers not existing). There are likely complex geopolitical reasons for restoration site choice that go beyond “distance” and none of these social variables were included. This doesn’t need to be lengthy, but you need to more directly acknowledge the large amount of variation that remained unexplained in your model and what might be required in the future to identify the influential social-ecological factors at play here. Numerous papers discuss the strong influence of social factors in dictating reef structure and function in recent decades (e.g. Norstrom et al. 2016, Hughes et al. 2017, Williams et al. 2019), and it is highly likely it is some of these social variables driving site choice.

RESPONSE #60: See Response #52; > 40% model performance is decidedly not ‘low’ in ecology. Regardless, we have improved model performance following the implementation of the parameter-calibration procedure. The cross-validation accuracy is now rather good, with an average true skill statistic = 0.48 (standard deviation = 0.18), and average false positive and negative error rates = 0.25 (0.17) and 0.27 (0.22). However, there are certainly many other factors besides those we included in the model that might affect the choice of restoration sites. Also following advice from Reviewer 2 (Responses #10, #17, #19, #20, #26, #42), we have now emphasized the importance of local social aspects, as well as of other potential drivers for which we did not account in the model (L300–310).

L30 – “comprehensive dataset”. This is vague – please give overall replication in abstract as this is central to the paper’s rigour.

RESPONSE #61: We have now included more information on the dataset.

L36 – “near future” – please quantify this statement.

RESPONSE #62: We now specify that the ‘near future’ refers to the middle of the century (L39-40).

L39 – “special attention to future climate conditions” – bit vague. You focus on ocean warming trends, so why not just say that or at least provide more context here as it is the final sentence of your abstract (and arguably, therefore, one of the most important sentences in your entire paper).

RESPONSE #63: We have now revised the final sentence of the Abstract to specify ocean warming trends and their implications, as suggested. The new version provides clearer context about the importance of addressing ocean warming when developing restoration guidelines, making the conclusion more aligned with the overall focus of the paper.

L45 – I would encourage you to also look at a much more recent paper on coral reef ecosystem services by Woodhead et al. (2019)

RESPONSE #64: We have added context regarding the changing dynamics of coral reef ecosystem services in the Anthropocene and the increasing importance of understanding the social-ecological interactions that underpin these services (L48–52):

“As coral reefs are modified in the Anthropocene, the traditional roles and functions of reefs will also likely change. Research suggests that the benefits we receive from coral reefs will increasingly depend on the interactions between social and ecological systems, and the emergence of novel service configurations might result from changing environmental and social dynamics (Woodhead et al. 2019).”

L45-46 – I think you need to more directly state “and local human stressors” here and give some key examples. Some good empirical examples (not review/opinion papers) of local stressors comprising reefs in the context of climate change/bleaching are as follows: Gove et al. (2023), Graham et al. (2015)

RESPONSE #65: We have now integrated Gove et al. (2023) as an example of the interaction between local anthropogenic stressors and global climate change, showing that local stressors such as pollution and overfishing can exacerbate the effects of climate-induced coral bleaching (L52–56):

“However, coral reefs are increasingly threatened by climate change and local human stressors, such as overfishing and pollution (Hughes et al. 2017). Reducing these local impacts can improve reef resilience and recovery following climate-induced bleaching events. For example, reefs with reduced land-sea impacts in Hawai’i demonstrated a greater probability of maintaining coral cover after marine heatwaves (Gove et al., 2023).”

L49-50 – Need more balanced argument here given the relatively small number of papers out there on this topic. I would draw your attention to Fox et al. (2021) who showed quite the opposite point to the one you are making

RESPONSE #66: We have revised the text to acknowledge both perspectives. While we have noted that selective mortality can modify the resilience of coral communities under certain conditions, we also emphasized that the general trend supports the idea that repeated stress events erode the ecological resilience of reefs. This revision

provides a more balanced and nuanced argument, reflecting the variability of coral responses to thermal stress (L57–61):

“Predicting the responses of reefs to future disturbance is complicated by the variability of coral responses to thermal stress...”

L64 – “local scale” – please quantify here what you mean by this, it’s subjective.

RESPONSE #67: We have clarified the term ‘local scale’ to refer to restoration actions at the country scale or finer (L80).

L73-74 – Again need to mention here that coral loss following heat stress is often exacerbated by local human stressors. Gove et al. (2023) and Graham et al. (2015) both relevant here again, but so is Donovan et al. (2021)

RESPONSE #68: We have now included the point that coral loss following heat stress is often exacerbated by local human stressors. We have also cited the relevant studies (see Response #65; L52–56).

L103-105 – Please elaborate on what you mean by “overall condition” and give some examples. I assume one is habitat complexity, which is often not re-gained even in restored systems (e.g. intertidal saltmarshes)

RESPONSE #69: We have now elaborated on what we mean by ‘overall condition’ and provided habitat complexity as an example. The revised text highlights that while coral survival is often measured as a success, factors such as habitat complexity, structural integrity, and biodiversity are also important for the long-term recovery of reef ecosystems (L145–148):

“This seemingly straightforward measure might be deceptive because of variable and often insufficiently long monitoring, and because the survival of the transplanted corals might not be representative of the overall condition of the entire reef, including habitat complexity, structural integrity, or biodiversity (Graham et al 2011).”

L106-110 – You don’t mention recent paper by Lange et al. (2024) about measuring carbonate budgets of “restored” reefs that seems highly relevant here as this at least tries to look in more detail at the habitat attributes of “restored” reefs. This paper is also relevant in L116-120.

RESPONSE #70: We have now added the reference in another context (see Response #17; L81–86):

L114-115 – It would seem relevant here to also mention that some agencies don't even actually monitor for success or not, see recent paper by Lamont et al. (2023)

RESPONSE #69: We have now added this citation to highlight the issue of insufficient monitoring by some agencies. Lamont et al. (2023) emphasized the need for greater rigour, transparency, and accountability in restoration, especially given the increasing involvement of large corporations (L167–170):

“Many restoration projects lack comprehensive monitoring frameworks to track long-term success. Even large transnational corporations that claim leadership roles in ecosystem restoration often fail to report on the outcomes of their restoration efforts (Lamont et al. 2023).”

L123 “220 restoration” – change to “220 coral restoration”. Also, we need a map of these localities. I found it frustrating that I had to go to the paper you cite in your methods to find a map. Please either provide as a Supp. Fig, or consider adding it to Figure 2 as an additional panel above.

RESPONSE #70: We did in fact include the restored localities in the maps in Fig. 6 (as black dots), but we concede that they were not easily visible. Also following the reviewer's comment below (Response #78), we now provide larger maps where the restored sites are more evident (Fig. 6). We have also produced another map showing only the restored locations, colour-coded by the year when the restoration was done (Extended Data Figure 1).

L148 – Why did you not use the “Gravity” metric previously developed for coral reefs by Cinner et al. (2018) which combines travel time with population size? This would seem a more refined method than an arbitrary cut-off population center size (see next comment).

RESPONSE #71: We have now added ‘gravity’ in the set of independent variables, which improved the models (L204). However, we also kept our original measure of remoteness in the analysis because it is not equivalent to nor is it correlated with gravity. We cannot reliably ascertain whether one method is more refined than the other. Instead, the two metrics convey different messages. Gravity represents a measure of human impact on reefs (higher gravity = more humans exerting pressure on a given reef locality), and remoteness (expressed as travel time from human settlements) is a measure of isolation, quantifying how difficult it is to reach a reef

location; remoteness is identical for localities at the same travelling distance from their closest human settlements, regardless of the size of the respective settlements. From the impact-exposure perspective, the two reefs would be different, but they would pose similar logistical challenges to the implementation and monitoring of restoration actions. In our original analyses, we were interested in this aspect precisely because we wanted to test the hypotheses that reefs posing fewer logistical challenges were more likely to be chosen as restoration targets.

L149 – “large” – please quantify. What was the threshold population density to be considered “large”?

RESPONSE #72: We have now clarified the definition of ‘large’ population density by adding a reference to the ‘high-density centres’ variant of the Global Human Settlement Layer (GHSL).

L168 – Figure 2, we need to know the replication along the x-axis of each partial dependency plot. You could do this by adding deciles (sometimes called “rug plots”) to the top of each plot. Also need x-axis label on panel “a”.

RESPONSE #73: Done.

L206 – “4 <” – shouldn’t this be the other way round “<4”?

RESPONSE #74: Corrected (L259).

L208-210 – Please also report the % that experienced more than 8 DHW as this is a common threshold that causes severe and widespread bleaching (sensu Skirving et al. 2020)

RESPONSE #75: Done (L261).

L215 – Is SSP2-4.5 the trajectory we are currently on? I didn’t think it was. Either way, please explain and justify to reader more why you chose this particular SSP projection.

RESPONSE #76: Actually, yes. The SSP2 scenario (“Middle of the road”) depicts a future where the world follows a path in which social, economic, and technological trends remain close to the historical patterns. It therefore represents a reasonable choice to explore future patterns without falling into either excessive pessimism (SSP3-5) or optimism (SSP1). We now provide a brief description of the narrative

behind the scenario and motivate its choice. We also added two relevant references (L269–273):

“Under an intermediate greenhouse gas-emissions scenario (SSP2-4.5) O’Neill et al. 2016; O’Neill et al. 217, where the world does not shift substantially from historical social, economic, and technological patterns — thus presenting moderate mitigation and adaptation challenges and an associated radiative forcing of 4.5 W m^{-2} — 90.9% of reef localities are predicted to experience at least one thermal anomaly/severe bleaching event by the end of the century (Fig. 5 and 6).”

L228 – please explain what “other reef localities” are in more detail in figure caption.

RESPONSE #77: We have updated the figure caption to clarify that ‘other reef localities’ refers to reef sites within the same geographic area that have not undergone restoration based on the dataset we used.

“Figure 5 | Number of predicted end-of-century thermal anomalies (\log_e) compared to the timing of the first anomaly under an intermediate-emissions scenario (SSP2-4.5). Black dots indicate restored localities; grey dots represent the other reef localities which have not undergone restoration ($0.5^\circ \times 0.5^\circ$ grid cells).”

L232 – Figure 6 – the maps are too small. Suggest they are made full page width and put the box plots below.

RESPONSE #78: Done.

L238 – change to “reef accessibility”.

RESPONSE #79: Done (L295).

L264 – a more appropriate term these days is “uncrewed” as it is gender neutral.

RESPONSE #80: We have now removed the sentence, so this no longer applies; but we note the term for future reference.

L271-273 – it would be useful context to refer some other database examples here that have been successful, for example Mermaid for reef monitoring: <https://datamermaid.org/>

RESPONSE #81: We have now incorporated a reference to the Mermaid platform as a successful example of a centralized, open-access database used for reef monitoring (L259–361):

“Thus, the database should accommodate data from diverse methodologies to provide a comprehensive resource for all types of coral restoration. Successful examples of similar databases include the Mermaid platform for reef monitoring (datamermaid.org), which provides an effective model for data standardization and sharing in marine science.”

This provides useful context for our proposal and highlights how similar systems can serve as models for coral restoration efforts.

L349 – Your choice of the term “environmental” here is perhaps misleading given you also include biological/ecological factors like coral diversity. I would suggest just using the term “factors” when referring to your predictor variables.

RESPONSE #82: Done (L467).

L516 – Latin name should be in italics.

RESPONSE #83: We have now edited the references accordingly (L699).

References cited in this response:

- Bayraktarov, E. et al. Motivations, success, and cost of coral reef restoration. *Restoration Ecology* 27, 981-991, doi:10.1111/rec.12977 (2019).
- Bellwood, D. R., Hughes, T. P., Folke, C. & Nystrom, M. Confronting the coral reef crisis. *Nature* 429, 827-833, doi:10.1038/nature02691 (2004).
- Breiman, L. Random Forests. *Machine Learning* 45, 5–32, doi: 10.1023/A:1010933404324 (2001)
- Brodie, J. et al. Chapter 28 - The Future of the Great Barrier Reef: The Water Quality Imperative. in *Coasts and Estuaries* (eds. Wolanski, E., Day, J. W., Elliott, M. & Ramachandran, R.) 477–499. doi:10.1016/B978-0-12-814003-1.00028-9. (Elsevier, 2019).
- Boström-Einarsson, L. et al. Coral restoration – A systematic review of current methods, successes, failures and future directions. *PLoS One* 15, e0226631, doi: 10.1371/journal.pone.0226631 (2020).
- Brown, K. T. & Barott, K. L. The Costs and Benefits of Environmental Memory for Reef-Building Corals Coping with Recurring Marine Heatwaves. *Integr. Comp. Biol.* 62, 1748–1755, doi: 10.1093/icb/icac074 (2022).
- Breiman, L. Random forests. *Machine Learning* 45, 5-32, doi:10.1023/A:1010933404324 (2001).
- Cinner, J. E. et al. Gravity of human impacts mediates coral reef conservation gains. *Proceedings of the National Academy of Sciences* 115, E6116-E6125, doi:10.1073/pnas.1708001115 (2018).

- Donovan, M. K. et al. Local conditions magnify coral loss after marine heatwaves. *Science* 372, 977-980, doi:10.1126/science.abd9464 (2021).
- Duarte, C. M. et al. Rebuilding marine life. *Nature* 580, 39–51. doi: 10.1038/s41586-020-2146-7 (2020).
- Fox, M. D. et al. Increasing Coral reef resilience through successive marine heatwaves. *Geophysical Research Letters* 48, e2021GL094128, doi:10.1029/2021GL094128 (2021).
- Friedman, J. H. Greedy function approximation: a gradient boosting machine. *The Annals of Statistics* 29, 1189-1232, doi:10.1214/aos/1013203451 (2001).
- Gove, J. M. et al. Coral reefs benefit from reduced land–sea impacts under ocean warming. *Nature* 621, 536-542, doi:10.1038/s41586-023-06394-w (2023).
- Graham, N. A. J., Nash, K. L. & Kool, J. T. Coral reef recovery dynamics in a changing world. *Coral Reefs* 30, 283–294, doi: 10.1007/s00338-010-0717-z (2011).
- Graham, N. A. J., Jennings, S., MacNeil, M. A., Mouillot, D. & Wilson, S. K. Predicting climate-driven regime shifts versus rebound potential in coral reefs. *Nature* 518, 94-97, doi:10.1038/nature14140 (2015).
- Greenwell, B. M., Boehmke, B. C. & McCarthy, A. J. A simple and effective model-based variable importance measure. *arXiv*, 1805.04755, doi:10.48550/arXiv.1805.04755 (2018).
- Hackerott, S., Martell, H. A. & Eirin-Lopez, J. M. Coral environmental memory: causes, mechanisms, and consequences for future reefs. *Trends Ecol. Evol.* 36, 1011–1023, doi: 10.1016/j.tree.2021.06.014 (2021).
- Halpern, B. S. et al. Recent pace of change in human impact on the world’s ocean. *Scientific Reports* 9, 11609, doi:10.1038/s41598-019-47201-9 (2019).
- Hughes, T. P. et al. Coral reefs in the Anthropocene. *Nature* 546, 82-90, doi:10.1038/nature22901 (2017).
- Hughes, T. P. et al. Global warming and recurrent mass bleaching of corals. *Nature* 543, 373-377, doi:10.1038/nature21707 (2017).
- Hughes, T. P., Baird, A. H., Morrison, T. H. & Torda, G. Principles for coral reef restoration in the anthropocene. *One Earth*, doi : 10.1016/j.oneear.2023.04.008 (2023).
- Kleypas, J. et al. Designing a blueprint for coral reef survival. *Biol. Conserv.* 257, 109107, doi: 10.1016/j.biocon.2021.109107 (2021)
- Lamont, T. A. C. et al. Hold big business to task on ecosystem restoration. *Science* 381, 1053-1055, doi:10.1126/science.adh2610 (2023).
- Lange, I. D. et al. Coral restoration can drive rapid reef carbonate budget recovery. *Current Biology* 34, 1341-1348.e1343, doi:10.1016/j.cub.2024.02.009 (2024).
- Lawrence, P. J., Smith, G. R., Sullivan, M. J. P. & Mossman, H. L. Restored saltmarshes lack the topographic diversity found in natural habitat. *Ecological Engineering* 115, 58-66, doi: 10.1016/j.ecoleng.2018.02.007 (2018).
- Müller, D., Leitão, P. J. & Sikor, T. Comparing the determinants of cropland abandonment in Albania and Romania using boosted regression trees. *Agricultural Systems* 117, 66-77, doi: 10.1016/j.agsy.2012.12.010 (2013).
- Nalley, E. M. et al. Water quality thresholds for coastal contaminant impacts on corals: A systematic review and meta-analysis. *Sci. Total Environ.* 794, 148632, doi: 10.1016/j.scitotenv.2021.148632 (2021).
- Norström, A. V. et al. Guiding coral reef futures in the Anthropocene. *Frontiers in Ecology and the Environment* 14, 490-498, doi:10.1002/fee.1427 (2016).
- Núñez Lendo, C. I. et al. Carbonate budgets induced by coral restoration of a Great Barrier Reef site following cyclone damage. *Frontiers in Marine Science* 10, 1298411, doi:10.3389/fmars.2023.1298411 (2024).
- O’Neill, B. C. et al. The Scenario Model Intercomparison Project (ScenarioMIP) for CMIP6. *Geosci. Model Dev.* 9, 3461–3482, doi:10.5194/gmd-9-3461-2016. (2016).
- O’Neill, B. C. et al. The roads ahead: Narratives for shared socioeconomic pathways describing world futures in the 21st century. *Glob. Environ. Change* 42, 169–180, doi: 10.1016/j.gloenvcha.2015.01.004 (2017).
- Pesaresi, M. & Freire, S. Global Human Settlement. European Commission. <https://human-settlement.emergency.copernicus.eu/download.php> (2023)
- Skirving, W. et al. CoralTemp and the Coral Reef Watch Coral Bleaching Heat Stress Product Suite Version 3.1. *Remote Sensing* 12, doi:10.3390/rs12233856 (2020).

- Scott, R. I. et al. Cost-effectiveness of tourism-led coral planting at scale on the northern Great Barrier Reef. *Restoration Ecology* 32, e14137, doi:10.1111/rec.14137 (2024).
- Shaver E C, Courtney C A, West J M, Maynard J, Hein M, Wagner C, Philibotte J, MacGowan P, McLeod I, Boström-Einarsson L, Bucchianeri K, Johnston L, Koss J. A Manager's Guide to Coral Reef Restoration Planning and Design. NOAA Coral Reef Conservation Program. NOAA Technical Memorandum CRCP 36, 120 pp., doi: 10.25923/vht9-tv39 (2020)
- Sing Wong, A., Vrontos, S. & Taylor, M. L. An assessment of people living by coral reefs over space and time. *Global Change Biology* 28, 7139-7153, doi:10.1111/gcb.16391 (2022).
- Strain, E. M. A. et al. A global assessment of the direct and indirect benefits of marine protected areas for coral reef conservation. *Divers. Distrib.* 25, 9–20, doi: 10.1111/ddi.12838 (2019).
- Streit, R. P. et al. Coral reefs deserve evidence-based management not heroic interference. *Nature Climate Change*, 14(8), 773-775, doi: 10.1038/s41558-024-02063-6 (2024)
- Suggett, D. J. & van Oppen, M. J. Horizon scan of rapidly advancing coral restoration approaches for 21st century reef management. *Emerg. Top. Life Sci.* 6, 125–136, doi: 10.1042/ETLS20210240 (2022).
- Suggett, D. J., Edwards, M., Cotton, D., Hein, M. & Camp, E. F. An integrative framework for sustainable coral reef restoration. *One Earth* 6, 666-681, doi: 10.1016/j.oneear.2023.05.007 (2023).
- Suggett, D. J. et al. Restoration as a meaningful aid to ecological recovery of coral reefs. *npj Ocean Sustainability* 3, 20, doi:10.1038/s44183-024-00056-8 (2024).
- Van Hooidek, R. et al. Local-scale projections of coral reef futures and implications of the Paris Agreement. *Sci. Rep.* 6, 1–8, doi: 10.1038/srep39666 (2016)
- Vardi, T. et al. Six priorities to advance the science and practice of coral reef restoration worldwide. *Restor. Ecol.* 29, e13498, doi: 10.1111/rec.13498 (2021).
- Williams, G. J. et al. Coral reef ecology in the Anthropocene. *Functional Ecology* 33, 1014-1022, doi:10.1111/1365-2435.13290 (2019).
- Woodhead, A. J., Hicks, C. C., Norström, A. V., Williams, G. J. & Graham, N. A. J. Coral reef ecosystem services in the Anthropocene. *Functional Ecology* 33, 1023-1034, doi:10.1111/1365-2435.13331 (2019).

Reviewer #1

No remarks.

Reviewer #2

Thank you to the Authors for taking the time to respond and edit their manuscript, in detail, to the points I raised in my first review. The paper reads much more balanced around the complexities that exist with reef restoration, and I happy to recommend publication in its current form. Well done again!

Thank you. No response required.

Reviewer #3

I appreciate all the work the authors have put into addressing my concerns/suggestions in the initial review of this work (I was the original Reviewer 3). The statistical modelling approaches are now far clearer and robust, and it is now much easier for the reader to follow the logic of the approaches. There are, however, a few remaining things to iron out, so that graphics are streamlined and convey the critical piece of information, and that key methods steps are clearly defended and justified, as well as a series of other minor suggested edits for clarity.

1. I still think the title of the paper should focus more on the results at hand, rather than the overall 'Discussion' point of the piece. For example: Coral reef restoration efforts are not guided by local human impacts or climate vulnerability or Coral reef restoration efforts are not guided by local and global human impacts. This is clearly something the editors can discuss with the authors.

RESPONSE #1: We understand the reviewer's opinion, and we are happy to discuss options with the editors. However, the current title is the third one we have proposed. Of course, this does not mean that the title is necessarily perfect, but our current title is the result of many rounds of internal discussion. Although our paper is a quantitative contribution based on data analysis and modelling, its goals and findings remain broad, and span multiple, complementary results. Our title, although not referring to a specific finding, reflects the take-home message of our work. In contrast, the titles suggested by the reviewer capture only the results from the first analysis (i.e., the one investigating the determinants of the choice of restoration sites), while they disregard the results on the determinants of restoration success, or future projections.

2. Line 56 – this is slightly inaccurate. Gove et al. focus on 'reef-builder' recovery following bleaching-induced coral mortality (i.e. hard coral + crustose coralline algae). Suggest you rephrase to "maintaining reef-builder cover (hard coral + CCA)".

RESPONSE #2: We have rephrased as suggested.

3. Line 66 – "out-plantation" – should this be "outplanting"?

RESPONSE #3: We have substituted “outplantation” with “outplanting”.

4. Line 82-84 - *Are these really “regional-scales”? This implies intra-country scales. I thought these examples were much smaller? I would just state the actual restoration scale of these examples, rather than trying to generalise (which could be misleading). For example, the Lange et al paper is talking about 4-8 ha area only: “To date, the Mars Coral Reef Restoration Program (buildingcoral.com), has restored >4 ha of reef across two neighboring islands in the Spermonde Archipelago, South Sulawesi (and >8 ha across Indonesia in total).”*

RESPONSE #4: We now clarify what we mean by local scale and larger scale, reporting the actual values as suggested (L79–84):

“Many early efforts suggest that restoration actions can succeed at the local scale (between 200 m² and 2 ha) and within a moderate time horizon (i.e., months to years) (Williams et al. 2019, Thornton et al. 2000, Kotb 2016, Rodgers et al. 2017). There are emerging examples where reef function has been successfully restored at a relatively larger scale (up to 8 ha across the country). For instance, recent studies have shown that coral restoration can drive rapid recovery of reef carbonate budgets following disturbances (Lange et al. 2024, Nuñez Lendo et al. 2024)”

5. Line 134 *“in specific occasions” is odd phrasing. Do you mean “under specific conditions”?*

RESPONSE #5: We have changed the phrase to “under specific conditions”.

6. Line 137 – *“such as reducing greenhouse gas emissions” – while I appreciate the importance of this, I do think the role played by local human land-sea impacts is also worthy of mention alongside this. Plenty of evidence of this driving reef decline (e.g. Gove et al. 2023).*

RESPONSE #6: We realize the importance of mentioning local human land-sea impacts, so we have changed this to:

“... such as reducing greenhouse-gas emissions and mitigating local human land-sea impacts ...” L115

7. Line 155 – *“outcomes” – perhaps “interpretations” is more suitable here? I don’t see how you can have an inaccurate outcome....the outcome is the outcome.*

RESPONSE #7: We have removed the sentence to shorten the MS.

8. Line 186 – *remoteness from what? Need to define from what early on to the reader at this point.*

RESPONSE #8: We now specify that we considered remoteness from large human settlements.

9. Line 204 – define “large” here please for the reader, rather than them having to look in methods as this represents crucial context.

RESPONSE #9: Because the definition is a bit technical (“contiguous localities with population density of at least 1,500 inhabitants per km², or more than 50% built-up density, and $\geq 50,000$ inhabitants”), we prefer to provide the details in the Methods.

10. Line 211 – “devised” – perhaps “built” is more appropriate?

RESPONSE #10: We have changed “devised” to “built” L189

11. Line 221 – “more pristine”. There are no “pristine” reefs left, whatever that might mean, so I think you’re safer just saying “less impacted” here.

RESPONSE #11: We have modified as suggested.

12. Figure 2 – while I appreciate your attempts to follow my request in the original review here, this is not what I was getting at, and I did in fact mean “ggPDdual_boot” function. This is confusing - what are the fitted trend lines in the ‘b’ panels then and why do they differ from in ‘a’ (noting you don’t explain what these are clearly in the figure legend)? My original suggestion was to add the fitted values to your fitted functions (i.e. add fitted values to your ‘a’ panels). As mentioned in my previous review, this can be done using the ggPDdual_boot function (I did not mean the ggPDfit function as you suggest in your response). We need the following three pieces of information all on each of the partial dependency plots: 1) fitted function (orange line in example figure below), 2) error around the fitted function (grey shading), 3) underlying fitted values (grey open circles) - see below example we built for an analysis. The ggPDdual_boot function will help you do this, as I’m sure there are other functions perhaps by now too. Your PDfit plot is not useful, as it’s unclear what the fitted function in this represents (and why it is different from the fitted function from your model outputs in panels ‘a’). Attached is one we built for some drivers of water quality analyses. Note how each partial dependency plot has the 3 pieces of information I mention above on each plot (hopefully this helps clarify).

RESPONSE #12: We think the confusion here originates from the solid lines and shaded areas in the fitted-values plots. These do not represent fitted functions; they are simply smoothers for the fitted values. That is why they differ from the fitted functions in the top panels. We have now redrawn the plots showing the fitted function and the fitted values together as requested, reporting on the y axes the predicted probability of restoration site selection.

We could not find any function called ggPDdual_boot in the ggBRT package. We think the reviewer is referring to the function ggPD_boot, which provides a bootstrapped version of the fitted functions obtained with the function ggPD. However, we went through the code and documentation of the ggBRT package again, and we realize that some details of the functions were not well-documented (the package is not in CRAN). So, in the end, we

decided to draw the plots from scratch using the output from the partial function from the pdp package.

Figure 2 | Factors affecting choice of restoration sites. Partial dependency plots showing the marginal effect of the independent variables included in our model on the predicted probability of a site to be the target of restoration. We report both the fitted functions (solid orange lines), and the fitted values (blue dots). Rugs on top of panels show the density of observed values in the target independent variables. The percentage values in parentheses indicate the relative influence of each variable.

13. Line 234 – “technique/s” – should be “technique(s)”

RESPONSE #13: Done

14. Line 240 - It's not clear to me how reference to Extended Data Figure 4 here helps this part of the story here. Please help the reader see the links more.

RESPONSE #14: Extended Data Figure 4 supports the explanation of how we calculated success in relation to the duration of post-restoration monitoring.

“... assessed through cross validation; see Methods – *Determinants of short-term restoration success* and Extended Data Figure 4 for details on how restoration success was assessed.”

15. Figure 3 – I don't see you refer to this figure anywhere in the main text. Please explain/fix. Need to help the reader understand why this plot is relevant to the story more in the main text.

RESPONSE #15: We now refer to Figure 3 in L208.

16. Line 268-269 - But can't an increase of ~1degC trigger bleaching depending on the number of DHW? I do not understand the motivation for this conservative approach. It would be useful to see

both 1.0 (or 1.5) and 2.0 deg C (as a 2 x 2 panel plot), or more clearly defend/justify going against the standard NOAA protocol here.

RESPONSE #16: Due to the obvious uncertainties in future projections, we opted for the 2° C threshold as a conservative choice (as we also did in previous published work; see for instance Strona et al. 2021). However, we have now replicated the analyses and redrawn Fig. 5 and Fig. 6 by using DHW projections from “Mellin, C. et al. Cumulative risk of future bleaching for the world’s coral reefs | Science Advances. 10, (2024)”, identifying all the localities/year where the maximum projected DHW is ≥ 20 .

Figure 5 | Number of predicted end-of-century coral mass mortality years compared to the timing of the first mass mortality year under an intermediate-emissions scenario (SSP2-4.5). Black dots indicate restored localities; grey dots represent the other reef localities that have not undergone restoration ($0.5^\circ \times 0.5^\circ$ grid cells). Mass mortality years for a given locality are identified as those where the maximum projected degree heating week (DHW) was ≥ 20 . The analysis is based on projected DHW data from ref⁸².

Figure 6 | First occurrence and frequency of years when reef localities experienced a maximum degree heating week (DHW) ≥ 20 under an intermediate-emissions scenario (SSP2-4.5). Restored sites identified in the maps by small black dots within the coloured pixels. Projected DHW data are from ref⁸².

17. Line 274-275 - Ok, but it's more about the frequency here than the average timing of the anomaly isn't it? Reefs can cope with being bleached (they have adapted to), but not if it happens too often. We want to know if anomalous temperatures will occur at such a frequency as to compromise recovery windows. This would be much more informative if expressed as 'number of anomalies per decade'. It could then be plotted as a continuum rather than just quoting these mean timing values. See McWhorter et al. 2021 (Fig. 1b) for inspiration here: onlinelibrary.wiley.com/doi/epdf/10.1111/gcb.15994

RESPONSE #17: We do in fact report that information (i.e., the frequency of anomalies) on top of the timing of the first anomaly. We reported that as total number of predicted bleaching events by the end of the century. This information, combined with the map showing the timing of the first anomaly, provides a comprehensive overview of which localities will be more affected by frequent bleaching events in the future, and when. We could provide small, individual maps showing the number of bleaching events per decade in a multi-panel figure, but that would be a bit overwhelming and not particularly informative. It would also go against previous criticism/suggestions by the reviewers on the need to provide large maps to ease visualization.

The figure indicated by the reviewer shows global temporal trends of anomalies/bleaching events under different future emissions scenarios. We have now added an analogous figure in the Supplementary Information (Extended Data Figure 5).

Extended Data Figure 5 | Average number of sites experiencing a maximum yearly degree heating week (DHW) ≥ 20 per decade. The plot shows the average number of sites with predicted maximum yearly DHW ≥

20 globally within a moving time window of 10 years. We mapped reef localities at a resolution of $0.5^\circ \times 0.5^\circ$ latitude/longitude ($n = 3780$). The moving window started at year 2015, so the first reported year is 2025. Predictions are based on an intermediate CMIP6 climate projection (SSP2-4.5, “middle of the road”; see Methods for details), with DHW obtained from ref⁸².

18. Line 281 - *Why do titles on graph indicate “I-II” and “II” ...is this supposed to mean I to II and than II or greater? Not clear.*

RESPONSE #18: “I-II” means that localities were exposed to bleaching alert levels I and/or II. “II” means they were only exposed to bleaching level II.

19. Line 304-306 - *This question has be raided before and discussed in detail (please acknowledge this work: [conbio.onlinelibrary.wiley.com/doi/abs/10.1111/j.1523-1739.2008.01037.x](https://onlinelibrary.wiley.com/doi/abs/10.1111/j.1523-1739.2008.01037.x)*

RESPONSE #19: We now cite the suggested reference.

20. Line 237-238 - *This is a little too brief - please expand slightly for the reader. Observed inconsistency in what? You mean the poor performance of your model?*

RESPONSE #20: L266–268:

“Differences in coral species targeted and the extent of local anthropogenic impacts also likely contributed to the poor performance of the model, complicating the identification of predictors of success.”

21. Line 377-379 - *You cite 2 review papers here. Better to cite primary data papers for these kinds of statements, for example this paper that you already cite in Intro: agupubs.onlinelibrary.wiley.com/doi/full/10.1029/2021GL094128*

RESPONSE #21: We agree that citing data papers is important and have included it with the two review papers.

22. Line 416 - *The Gove et al. paper is very relevant again here as this shows you only achieve reef resilience to bleaching if both land and sea based stressors are mitigated simultaneously.*

RESPONSE #22: We have revised the text to highlight the importance of addressing both land- and sea-based stressors simultaneously to achieve reef resilience to bleaching, as emphasized in the Gove et al. paper. L347–350:

“For instance, combined approaches targeting both sea- and land-based stressors (Gove et al. 2023) such as marine protected areas (Strain et al. 2019) and water-quality remediation (Nalley et al. 2021, Brodie et al. 2019) could create synergistic benefits for reefs, while also supporting local communities with incentives for conservation.”

23. Line 486-488 - This is all much clearer now. However, these bag fractions of 0.8 are high. How does your model performance vary if you drop this down in increments of 0.1 to 0.5? I would prefer that you report the range in model performance values having done this sensitivity analysis. I know this is work, but I think that will be more reassuring to the reader and prevent folks worrying about overfitting here because of a high proportion of training data.

RESPONSE #23: We are unsure about this suggestion. The choice of the 80/20 ratio in training/testing set size is consistent with recommendations and common practices in machine learning. See for instance Tan et al. (2021), Vrigazova (2021), and Sivakumar et al. (2024).

Increasing the size of the training set is usually considered a good strategy to increase the model’s ability to generalize, because it results in exposing the algorithm to a wider amount of information to identify patterns in new data. Clearly, having a large training set obviously implies having a small testing set to evaluate model accuracy.

However, that ratio might result in a biased evaluation of model accuracy only if there are biases in the quality of the data included in the testing set; i.e., if they tend to over-represent some specific feature of the data in the training set, or if they are not independent from the training data (e.g., because of spatial autocorrelation). In our analyses, neither of the two issues is present because: (1) we replicated the cross-validation 1000 times, thereby ensuring that we tested the model across replicates on a wide range of test data, and (2) because we had already pre-processed the complete dataset to ensure full spatial independence between training and testing set.

This said, we tested model accuracy by replicating the cross-validation using a 50/50 ratio between the size of training and testing set, obtaining an average true skill statistic (TSS) = 0.4 (i.e., not much lower than the TSS of the models trained on 80% of the available data, i.e., 0.48). We report this result in L442–445:

“We also replicated this step by generating training sets including 50% of observations instead of 80%. The substantial reduction in the amount of training information provided to the model resulted in a small reduction of model accuracy, yielding an average TSS of 0.40 ± 0.12 (\pm standard deviation).”

24. Line 488-489 - Please re-phrase for non-expert. What is the issue you are worried about and what is the goal you are hoping to achieve with this approach? You simply report the 'method' without explaining the reasoning behind it.

RESPONSE #24: Rephrased; L419–423:

“To address the class imbalance in the training data, where restored sites were far less common than non-restored sites, we applied site weights inversely proportional to the prevalence of each class. This approach ensured that the less-common restored sites contributed more to the model training, while non-restored sites were downweighted to prevent the model from becoming biased toward the majority class.”

25. Line 510 – “relative influence” - Please at least reference the used methods approach here (I appreciate you do not need to add detailed formula).

26. Line 510 – “marginal variable effects” - Aren't these conditional effects? i.e. the effect of a given predictor holding all other predictors at their mean values?

RESPONSE #25 and #26: Rephrased; L447–448. We assessed partial dependency as in Friedman's original formulation (2001), also described in the `pdp` package documentation (journal.r-project.org/archive/2017/RJ-2017-016/RJ-2017-016.pdf):

Constructing a PDP (3) in practice is rather straightforward. To simplify, let $z_s = x_1$ be the predictor variable of interest with unique values $\{x_{11}, x_{12}, \dots, x_{1k}\}$. The partial dependence of the response on x_1 can be constructed as follows:

1. For $i \in \{1, 2, \dots, k\}$:
 - (a) Copy the training data and replace the original values of x_1 with the constant x_{1i} .
 - (b) Compute the vector of predicted values from the modified copy of the training data.
 - (c) Compute the average prediction to obtain $\bar{f}_1(x_{1i})$.
2. Plot the pairs $\{x_{1i}, \bar{f}_1(x_{1i})\}$ for $i = 1, 2, \dots, k$.

27. Lines 557-559 - Again - I think you need to make the motivation behind your approach/choice here very clear - why go against the standard NOAA approach? This will come as a surprise to readers and we need to know why you've deviated (you need to defend/justify this more I think).

RESPONSE #27: See Response #16.

References cited in this response document

Brodie, J. *et al.* Chapter 28 - The Future of the Great Barrier Reef: The Water Quality Imperative. in *Coasts and Estuaries* (eds. Wolanski, E., Day, J. W., Elliott, M. & Ramachandran, R.) 477–499 (Elsevier, 2019). doi:10.1016/B978-0-12-814003-1.00028-9.

- Friedman, J. H. Greedy function approximation: a gradient boosting machine. *Annals of statistics*, 1189-1232 (2001).
- Gove, J. M. et al. Coral reefs benefit from reduced land–sea impacts under ocean warming. *Nature* 621, 536–542 (2023).
- Kotb, M. M. A. Coral translocation and farming as mitigation and conservation measures for coastal development in the Red Sea: Aqaba case study, Jordan. *Environ. Earth Sci.* 75, 439 (2016).
- Lange, I. D. et al. Coral restoration can drive rapid reef carbonate budget recovery. *Curr. Biol.* 34, 1341-1348.e3 (2024).
- Nalley, E. M. et al. Water quality thresholds for coastal contaminant impacts on corals: A systematic review and meta-analysis. *Sci. Total Environ.* 794, 148632 (2021).
- Núñez Lendo, C. I. et al. Carbonate budgets induced by coral restoration of a Great Barrier Reef site following cyclone damage. *Front. Mar. Sci.* 10, (2024).
- Pesaresi, M. & Freire, S. Global Human Settlement. European Commission (2023).
- Rodgers, K. S. et al. Effectiveness of coral relocation as a mitigation strategy in Kāneʻohe Bay, Hawaiʻi. *PeerJ* 5, e3346 (2017).
- Sivakumar, M., Parthasarathy, S. & Padmapriya, T. Trade-off between training and testing ratio in machine learning for medical image processing. *PeerJ Comp. Sci.* 10, e2245, doi:10.7717/peerj-cs.2245 (2024).
- Strain, E. M. A. et al. A global assessment of the direct and indirect benefits of marine protected areas for coral reef conservation. *Divers. Distrib.* 25, 9–20 (2019).
- Strona, G. et al. Global tropical reef fish richness could decline by around half if corals are lost. *Proc. R. Soc. B* 288, 20210274 (2021).
- Tan, J., Yang, J., Wu, S., Chen, G. & Zhao, J. A critical look at the current train/test split in machine learning. *arXiv*, doi:10.48550/arXiv.2106.04525 (2021).
- Thornton, S. L., Dodge, R. E., Gilliam, D. S., DeVictor, R. & Cooke, P. Success and Growth of Corals Transplanted to Cement Armor Mat Tiles in Southeast Florida: Implications for Reef Restoration. in *Proceeding 9* vol. 2 23–27 (Nova Southeastern University, Ministry of Environment, Indonesian Institute of Sciences and International Society for Reef Studies, Bali, Indonesia, 2000).
- Vrigazova, B. The proportion for splitting data into training and test set for the bootstrap in classification problems. *Bus. Syst. Res. J.* 12, 228-242, doi:10.2478/bsrj-2021-0015 (2021).
- Williams, S. L. et al. Large-scale coral reef rehabilitation after blast fishing in Indonesia. *Restor. Ecol.* 27, 447–456 (2019).